# Adipocyte p53 coordinates the response to intermittent fasting by regulating adipose tissue immune cell landscape

Isabel Reinisch[1,2], Helene Michenthaler[1], Alba Sulaj [3,4], Elisabeth Moyschewitz[1], Jelena Krstic [1], Markus Galhuber[1], Ruonan Xu[1], Zina Riahi [1], Tongtong Wang[2], Nemanja Vujic [5], Melina Amor [5], Riccardo Zenezini Chiozzi[6], Martin Wabitsch [7], Dagmar Kolb[1,8], Anastasia Georgiadi [3], Lisa Glawitsch [9], Ellen Heitzer [9], Tim J. Schulz [10,11,12], Michael Schupp [13], Wenfei Sun[14], Hua Dong[15], Adhideb Ghosh [2,16], Anne Hoffmann [17], Dagmar Kratky [5,18], Laura C. Hinte [19], Ferdinand von Meyenn [19], Albert J. R. Heck [6], Matthias Blüher[20], Stephan Herzig[3,4], Christian Wolfrum [2] & Andreas Prokesch [1,18] ✉

In obesity, sustained adipose tissue (AT) inflammation constitutes a cellular memory that limits the effectiveness of weight loss interventions. Yet, the impact of fasting regimens on the regulation of AT immune infiltration is still elusive. Here we show that intermittent fasting (IF) exacerbates the lipid-associated macrophage (LAM) inflammatory phenotype of visceral AT in obese mice. Importantly, this increase in LAM abundance is strongly p53 dependent and partly mediated by p53-driven adipocyte apoptosis. Adipocyte-specific deletion of p53 prevents LAM accumulation during IF, increases the catabolic state of adipocytes, and enhances systemic metabolic flexibility and insulin sensitivity. Finally, in cohorts of obese/diabetic patients, we describe a p53 polymorphism that links to efficacy of a fasting-mimicking diet and that the expression of p53 and TREM2 in AT negatively correlates with maintaining weight loss after bariatric surgery. Overall, our results demonstrate that p53 signalling in adipocytes dictates LAM accumulation in AT under IF and modulates fasting effectiveness in mice and humans.

Obesity represents a major health problem, increasing the risk of mortality due to its comorbidities like type 2 diabetes, cardiovascular disease, depression, Alzheimer's disease, osteoarthritis, and cancer[1]. During obesity progression, the storage capacity of adipocytes gets exhausted, eliciting adipocyte stress and sterile inflammation in white adipose tissue (WAT)[2]. In detail, the cellular landscape of AT shifts towards an increased abundance of AT macrophages (ATMs), which constitute the predominant immune cell type in obese WAT. The switch from immunoregulatory to inflammatory signalling in WAT contributes to tissue dysfunction, driving the development of obesity-associated diseases[3]. While dietary interventions such as fasting regimens and surgical interventions like bariatric surgery constitute effective weight loss measures in combating the obesity epidemic[4,5], they are largely insufficient to restore AT function in mice[6–11] and humans[11–16]. The extent of the inflammatory fingerprint left from obesity correlates positively with weight regain after dietary interventions[9] and might explain the persistent risk for metabolic disease in formerly obese individuals[17,18].

Recent advances in single cell and nuclei RNA sequencing approaches greatly improved our understanding of cellular

maladaptation's in AT dysfunction and outdated the historical notion that only two macrophage subpopulations (M1 as pro-inflammatory and M2 as anti-inflammatory macrophages) exist[19,20]. In contrast, AT compromises a plethora of different macrophage subtypes, which flexibly adapt to different metabolic challenges[20–22] thereby dictating AT remodelling. Thus, it is essential to unravel weight loss-induced changes in the composition of AT to identify the relevant cell types as well as the intracellular mechanisms and intercellular communications that maintain an obesogenic memory in AT. Several hallmarks of dysfunctional adipocytes were described to depend on transcriptionally regulated mechanisms, such as apoptosis, senescence, or altered adipokine and lipokine secretion[3]. In this context, earlier studies in mouse models of obesity indicated an involvement of the transcription factor p53 in regulating adipocyte stress responses that might signal to recruit macrophages[23]. Additionally, akin to the response to increased energy intake, nutrient deprivation has been shown as a potent inducer of p53 signalling in various cell types by upstream cues like AMPK signalling, fatty acids, or DNA damage[24–27].

Here, by deconvoluting the cellular landscape of AT using single-nuclei RNA sequencing, we demonstrate that lipid-associated macrophages (LAMs) emerge in visceral AT in response to intermittent fasting (IF)-induced weight loss in obese mice in an adipocyte p53-dependent manner. Induction of p53 knock out in AT prevents LAM emergence and elevates the catabolic state of adipocytes, enhancing the systemic fasting reponse. Furthermore, we show that p53 status is associated with the outcome of dietary and surgical interventions in humans, thus representing a possible modulator to boost the long-term effectiveness of weight loss strategies.

## Results

### IF increases the abundance of crown-like structures in AT

To study AT plasticity in response to fasting cues, we induced obesity in mice by feeding them a 60% high-fat diet (HFD) for 12 weeks before challenging them with a 4:3 cyclic IF regimen (consisting of 3 fasting days and 4 feeding days per week, 24 h each) for 18 days (Supplementary Fig. 1a). As reported before (e.g. ref. 28), this protocol robustly elicited progressive weight loss (Fig. 1a), culminating in significantly reduced body weight and adiposity at the end of the experiment (Fig. 1b, c), when mice were still in the weight loss phase. Furthermore, systemic insulin sensitivity was improved by IF, albeit not reaching the levels of lean chow-fed mice, as analysed by glucose and insulin tolerance tests (Fig. 1d, e and Supplementary Fig. 1b, c). Weight loss and metabolic improvements occurred despite unchanged cumulative food intake (Fig. 1f), as IF mice compensated for time-restricted food availability within the first 4 h of refeeding (Supplementary Fig. 1d). Histological analysis of epididymal white adipose tissue (eWAT) depots revealed a substantial increase in the number of crown-like structures (CLS) in response to IF (Fig. 1g, h), which corresponded to a significant increase in mRNA expression of the pro-inflammatory marker Cd11c in eWAT (Fig. 1i). There were no overt changes in the expression of other M1 (Tnfα, Il6, Nos2, Ccl2) or M2 (Arg1) macrophage markers (Fig. 1j), indicating that the fasting-recruited macrophages fail to comply with the M1 vs. M2 classification paradigm. Recently, a novel subpopulation of Cd11c-positive macrophages has been identified and defined as LAMs[29]. Interestingly, transcript levels of genes encoding for LAM markers like Mmp12, Trem2, Lipa, and Gpnmb showed a clear trend to increased expression in response to IF when measured in bulk eWAT (Fig. 1k). Although the gene expression levels of the LAM markers Trem2 and Mmp12 were significantly lower in subcutaneous WAT (sWAT) and brown AT (BAT) compared to eWAT (Supplementary Fig. 1e), a certain degree of IF-mediated increase was found in all three adipose depots (Supplementary Fig. 1f, g). Furthermore, we detected increased expression of the LAM marker MMP12 on mRNA (Supplementary Fig. 1h) and protein level (Supplementary Fig. 1i) in response to an overnight fast also in eWAT of lean, healthy mice, suggesting that

fasting-induced LAM accrual is not dependent on prior established obesity.

Thus, these data suggest that (cyclic) fasting increases the abundance and/or alters the polarisation of macrophages in AT of lean and obese mice.

### IF elicits AT stress response via adipocyte-autonomous p53 signalling

The most potent mechanisms leading to macrophage recruitment in AT during obesity include stress-related pathways like apoptosis, senescence, and DNA-damage response[3]. We therefore hypothesised that adipocyte stress in response to IF is the primary trigger for macrophage infiltration. In line with this, we observed that inflammatory hotspots under IF are often located around small adipocytes (Fig. 1g), which were previously identified as dying adipocytes[30]. Moreover, despite a significant decrease in eWAT weight in response to IF (Fig. 1c), we could not detect a difference in average epididymal adipocyte size (Fig. 2a), suggesting that adipocyte death may contribute to the reduction in depot volume. In addition, the expression levels of several genes associated with apoptosis were upregulated in the adipocyte-rich fraction isolated from eWAT or sWAT of intermittently fasted (HFD-IF) in comparison to ad libitum fed (HFD-AL) obese mice, while markers of senescence (p16) and DNA-damage (Parp1) were not changed (Fig. 2b, c and Supplementary Fig. 2a measured in bulk tissue). Importantly, we detected a strong trend for increased abundance of cleaved caspase 3-positive cells in eWAT of obese IF mice (Fig. 2d, e). Hence, in association with an increased abundance of macrophages in AT, IF seems to evoke an adipocyte-autonomous apoptotic stress response. The transcription factor p53 is well-known to control cellular fate by dictating major stress-related pathways like apoptosis[31]. Previous findings demonstrated that p53 is activated in AT of obese and diabetic mice and humans in response to DNA damage, consequently inducing adipocyte senescence and the recruitment of macrophages[23]. Thus, we sought to test whether p53 signalling is induced and activates stress-related apoptosis in adipocytes upon nutrient withdrawal. We found that the gene expression of Trp53 and the p53 targets Cdkn1a and Mdm2 were upregulated in the adipocyte-rich fraction isolated from eWAT (Fig. 2f) and sWAT (Fig. 2g) in response to IF in HFD-fed mice. This fasting-evoked activation of the p53 signalling pathway was specific to AT depots, as the effect was absent in eWAT-derived stromal vascular fraction (SVF; Supplementary Fig. 2b), skeletal muscles (Supplementary Fig. 2c, d), and liver (Supplementary Fig. 2e). In liver we observed a significant reduction of hepatic lipid droplets in the HFD-IF compared to the HFD-AL group (Supplementary Fig. 2f), likely contributing to the improvements in systemic metabolism upon IF (Fig. 1).

To test whether the observed p53 and apoptotic induction upon nutrient withdrawal is cell-autonomous, we employed in vitro adipocyte models. Adipocytes differentiated from the SVF of AT of chow-fed wild-type mice, showed a starvation-induced p53 protein upregulation (Fig. 2h), concomitant with target gene induction to a level comparable to pharmacological p53 activation (Fig. 2i). p53 protein upregulation in starvation conditions was also observed in murine adipocytes differentiated from C3H10T1/2 cells (Fig. 2j). This was consistent with p53 target gene induction upon starvation, which was blunted upon p53 knock out using CrispR/Cas9 (Fig. 2k, l). Upon repletion of nutrient-rich culture medium, p53 protein and target gene expression levels were returned to basal levels after 3 h (Supplementary Fig. 2g, h), indicating reversibility of starvation-induced p53 signalling activation. In human adipocyte SGBS cells, we also observed an increase in the expression of p53 and p53-dependent target genes under starvation conditions that was attenuated after p53 knock-down with siRNAs (Fig. 2m). Genes encoding apoptotic proteins were also regulated in a p53-dependent manner (Fig. 2n). Globally, RNA sequencing of these SGBS adipocytes under starvation conditions yielded p53 knock

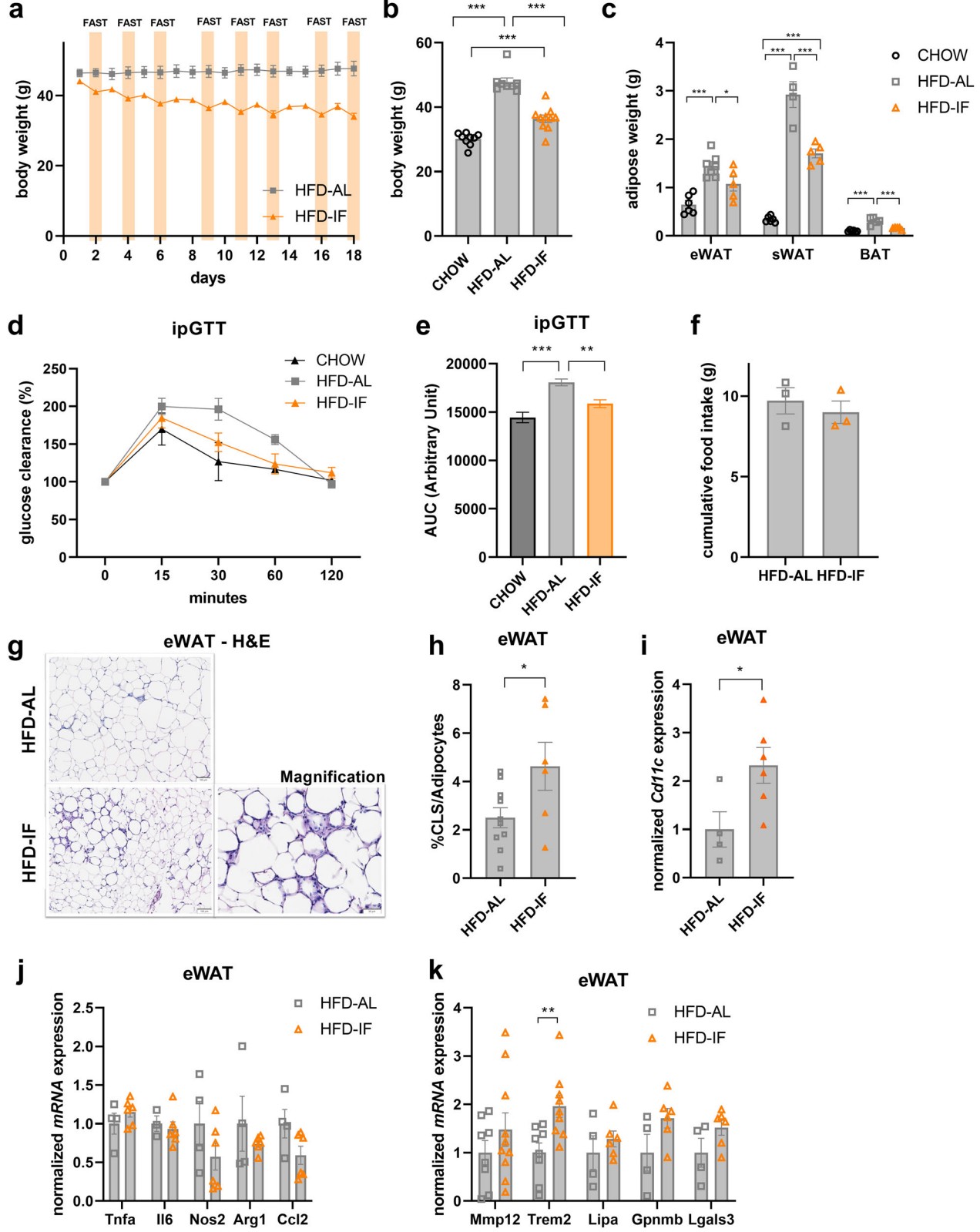

down-mediated de-enrichment in p53, apoptosis, and inflammatory response pathways (Supplementary Fig. 2i).

Together, these findings indicate that the p53 signalling pathway is induced in an adipocyte-autonomous manner in response to nutrient reduction, and might control the apoptotic fate and inflammatory response of adipocytes under metabolically challenging conditions.

## Stress-induced p53 regulates eWAT remodelling in response to IF

To further investigate the regulatory role of adipocyte-p53 in AT remodelling in response to IF, we generated an adipocyte-specific, tamoxifen-inducible p53 KO mouse model. Tamoxifen was applied after inducing obesity in male adiponectin-CreERT2 x p53-lox/lox (KO) mice or in control mice with p53-lox/lox lacking the Cre allele. After a

**Fig. 1 | Intermittent fasting increases the abundance of crown-like structures in adipose tissue (AT). a** Body weight of diet-induced obese mice of *ad libitum* controls (HFD-AL, *n* = 3 mice) and during the intermittent fasting intervention (HFD-IF, *n* = 4 mice; orange bars indicate 24 h water-only fasting). **b** Body weight of chow-fed lean (CHOW, *n* = 9 mice), HFD-AL (*n* = 8 mice), and HFD-IF (*n* = 10 mice) groups at the end of the experiment. **c** Adipose depot weights in grams of CHOW (*n* = 6 mice), HFD-AL (*n* = 4–7 mice), and HFD-IF (*n* = 5 mice) groups. eWAT epididymal white AT, sWAT subcutaneous white AT, BAT brown AT. **d, e** Intraperitoneal glucose tolerance test (ipGTT) of lean CHOW (*n* = 7 mice), HFD-AL (*n* = 6 mice), and HFD-IF (*n* = 13 mice) groups. Basal glucose levels are normalised and glucose clearance is shown in percent 15, 30, 60, and 120 min after glucose injection. **f** Cumulative food intake in grams of HFD-AL and HFD-IF mice (*n* = 3 mice per group), as analysed in metabolic cages. **g** Representative H&E stainings of eWAT of HFD-AL and HFD-IF mice. Scale bars are 100 μm or 50 μm for the magnification.

**h** Number of crown-like structures normalised to number of adipocytes in eWAT of HFD-AL (*n* = 10 mice) and HFD-IF (*n* = 6 mice) groups. **i, j, k** mRNA expression of *Cd11c* (**i**), of genes encoding for M1/M2 macrophage markers (**j**), and of lipid-associated macrophage markers (**k**) in eWAT of HFD-AL (*n* = 4–8 mice) and HFD-IF (*n* = 6–10 mice) mice. Data are presented as mean values ± SEM. Significant differences were analysed by two-tailed, unpaired *t*-test (**f**, **h–k**) or two-way (**a**, **d**) or one-way (**b**, **c**, **e**) ANOVA with Bonferroni post hoc tests. ****p* < 0.001, ***p* < 0.01, and **p* < 0.05. Exact *p* values: **b** CHOW vs. HFD-AL: <0.0001, HFD-AL vs. HFD-IF: <0.0001, CHOW vs. HFD-IF: 0.0007; **c** CHOW vs. HFD-AL: 0.0001, <0.0001, <0.0001, HFD-AL vs. HFD-IF: 0.0419, 0.0001, 0.0004, CHOW vs. HFD-IF: ns, <0.0001, ns (eWAT, sWAT, BAT respectively); **e** CHOW vs. HFD-AL: <0.0001, HFD-AL vs. HFD-IF: 0.0044; **h** 0.0371; **i** 0.0413; **k** *Trem2*: 0.0071. Source data are provided as a Source Data file.

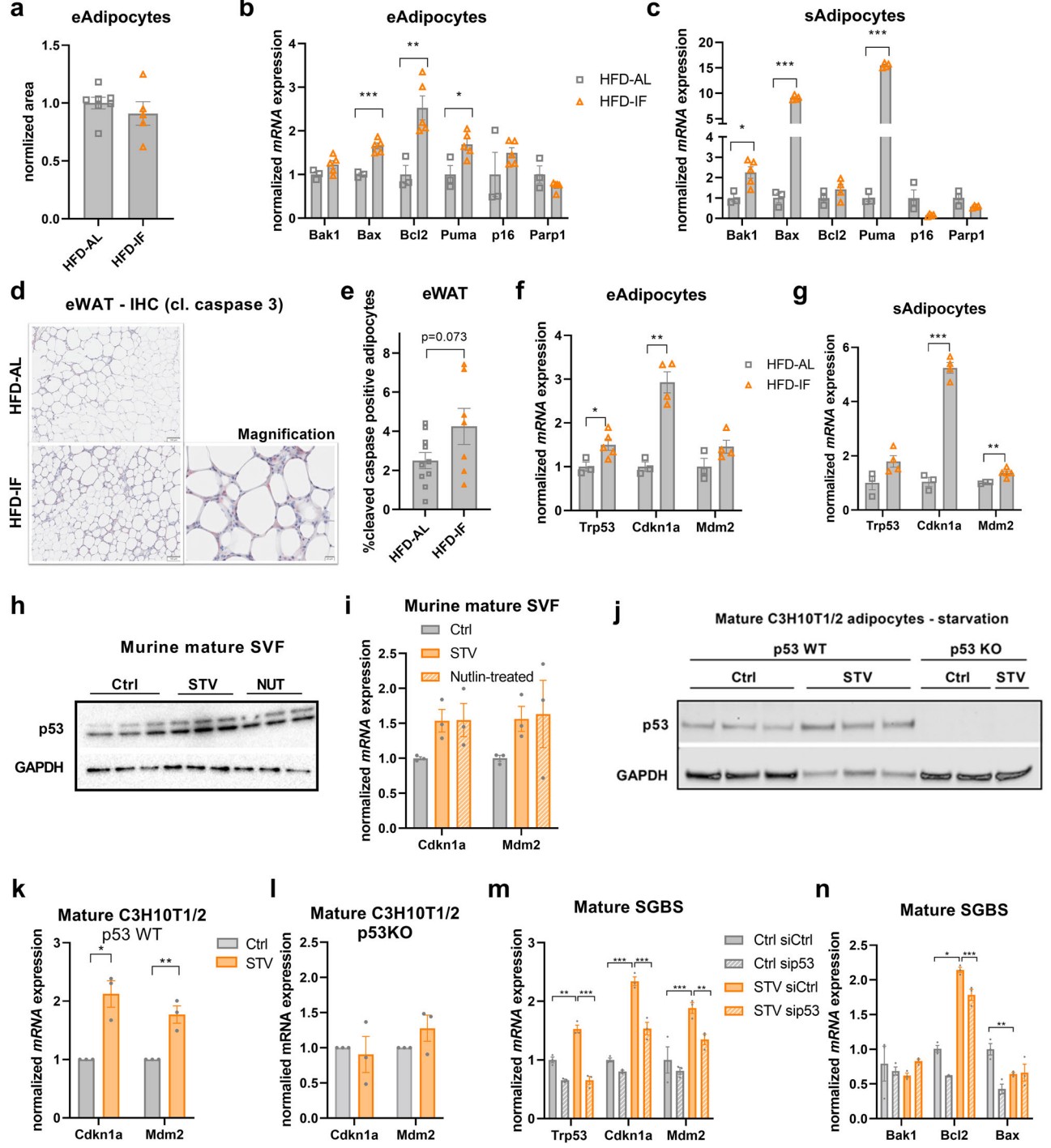

**Fig. 2 | IF elicits AT stress response via adipocyte-autonomous p53 signalling.**
**a** Cell size quantification from histology images showing the normalised area of epididymal adipocytes of HFD-AL (*n* = 7 mice) and HFD-IF (*n* = 5 mice) groups. **b**, **c** mRNA expression levels of genes encoding for apoptotic, senescence, or DNA damage-associated markers in the adipocyte-rich fraction isolated from eWAT (**b**) or sWAT (**c**) of HFD-AL (*n* = 3 mice) or HFD-IF (*n* = 5 mice) mice. **d** Representative images showing histological stainings of cleaved caspase 3 in eWAT of HFD-AL and HFD-IF mice. Scale bars are 100 μm or 20 μm for the magnification. **e** Quantification of cleaved-caspase positive adipocytes in eWAT of HFD-AL (*n* = 10 mice) and HFD-IF (*n* = 7 mice) mice. **f**, **g** mRNA expression of *Trp53* and p53 target genes *Cdkn1a* and *Mdm2* in the adipocyte-rich fraction isolated from eWAT (**f**) or sWAT (**g**) of HFD-AL (*n* = 3 mice) and HFD-IF (*n* = 5 mice). **h** Western blot analysis measuring p53 and GAPDH protein levels in isolated and differentiated cells from stromal vascular fraction (SVF) kept under nutrient-rich control (Ctrl) or starvation (STV) conditions or treated with 1 μM of idasanutlin (NUT) (*n* = 3 independent isolations). **i** mRNA expression levels of *Cdkn1a* or *Mdm2* in the differentiated SVF kept under Ctrl, STV or nutlin-treated conditions (*n* = 3 independent isolations). **j** Western blot analysis measuring p53 and GAPDH protein levels in differentiated p53 wildtype (WT) and

knock out (KO) C3H10T1/2 cells (*n* = 3 independent experiments) kept under nutrient-rich control (Ctrl) or STV conditions. **k**, **l** mRNA expression levels of *Cdkn1a* and *Mdm2* in p53 WT (**k**) and p53 knock out (KO, **l**) differentiated C3H10T1/2 cells kept under nutrient-rich control (Ctrl) or starvation (STV) conditions (*n* = 3 independent experiments). **m**, **n** mRNA expression levels of *Trp53*, *Cdkn1a*, and *Mdm2* (**m**) or apoptotic genes (**n**) in differentiated SGBS cells under nutrient-rich control (Ctrl) or STV conditions and treated with siRNA targeting p53 (sip53) or siRNA control (siCtrl) (*n* = 3 independent experiments). Data are presented as mean values ± SEM. Significant differences were analysed by two-tailed, unpaired *t*-test (**a**–**c**, **e**–**g**), one-sample *t*-test (**k**, **l**), or one-way ANOVA (**i**, **m**, **n**) with Bonferroni post hoc tests. ****p* < 0.001, ***p* < 0.01, and **p* < 0.05. Exact *p* values: **b** *Bax*: 0.0004, Bcl2: 0.0086, *Puma*: 0.0235; **c** *Bak1*: 0.0194, *Bax*: <0.0001, *Puma*: <0.0001; **f** *Trp53*: 0.0364, *Cdkn1a*: 0.0013; **g** *Cdkn1a*: <0.0001, *Mdm2*: 0.0086; **k** *Cdkn1a*: 0.0111, *Mdm2*: 0.0070; **m** Ctrl siCtrl vs. STV siCtrl: *Trp53*: 0.0025, *Cdkn1a*: <0.0001, *Mdm2*: <0.0001; STV siCtrl vs. STV sip53: *Trp53*: <0.0001, *Cdkn1a*: <0.0001, *Mdm2*: 0.0021; **n** Ctrl siCtrl vs. STV siCtrl: *Bcl2*: 0.0353, *Bax*: 0.0013; STV siCtrl vs. STV sip53: *Bcl2*: <0.0001. Source data and uncropped blots are provided as a Source Data file.

wash-out period of 1 week, control and p53 KO mice were submitted to the 4:3 IF regimen (Fig. 3a). Tamoxifen treatment caused a significant reduction in the expression of *Trp53* in adipocytes isolated from eWAT, sWAT, and in bulk BAT of CreERT2-positive mice, whereas the expression of *Trp53* in liver and skeletal muscle remained unaltered (Supplementary Fig. 3a). To obtain global insights into cellular plasticity in response to IF and in dependence of adipocyte p53 status, we performed single nuclei RNA sequencing (snRNA-seq) on eWAT of HFD-AL, HFD-IF, and HFD-IF-KO mice. Quality filtering and doublet removal yielded 12841, 14260, and 18238 single nuclei in the HFD-AL, HFD-IF, and HFD-IF-KO groups, respectively (Fig. 3b). The generated dataset enabled identification of the canonical gross cell types found in eWAT, including immune cells, adipocytes, fibro-adipogenic progenitors (FAPs), mesothelial cells, and endothelial cells (Fig. 3b), according to demarcation by their nuclear gene expression signatures (Fig. 3c). IF led to striking alterations in the cellular composition of AT, affecting almost all cell populations (Fig. 3d, e). Intriguingly, these changes in IF-mediated AT remodelling were largely absent in mice with adipocyte-specific p53 depletion (Fig. 3d, e) and were confirmed by a trending decrease in the number of CLS in histological sections (Supplementary Fig. 3b, c). Most notably, and in line with our measurements in bulk AT (Fig. 1g–i), immune cells were strikingly enriched in eWAT upon IF and this plasticity appeared to be largely controlled by p53 signalling in adipocytes (Fig. 3b, d, e). Similarly, the IF-mediated reduction of the adipocyte fraction (from 17% to ~8% under IF) was also dependent on p53, together indicating the potential involvement of p53-mediated cell death upon cyclic fasting as a recruiting signal for immune cells. To evaluate if acute p53 ablation in established obesity can rescue the inflammatory phenotype of AT, we analysed the expression of inflammatory markers in control and p53 KO mice before challenging them with IF (efficiency and specificity of ablation is shown in Supplementary Fig. 3d). The expression of *Tnfα*, *Adgre1*, *Nos2*, *Cd11c*, and *Trem2* in eWAT remained unaltered when comparing control and KO HFD-fed mice (Supplementary Fig. 3e), suggesting that cyclic fasting as an additional nutritional stressor is necessary to trigger p53-controlled macrophage accrual.

In snRNA-seq experiments, the variability between mice in one group is masked by pooling of eWAT depots of several mice. Thus, to support the validity of our snRNA-seq dataset, we performed untargeted proteomics on the same replicates that were pooled for snRNA-seq. In total, we detected about 6000 proteins, and principal component analysis correctly assigned the samples to the three experimental groups (Supplementary Fig. 3f). Hierarchical clustering identified five clearly distinct clusters of protein expression profiles (Fig. 3f). Overall, we detected 525 upregulated and 433 downregulated proteins in eWAT of HFD-IF mice compared to HFD-AL mice.

Importantly, a large proportion of these IF-mediated changes was dependent on the presence of p53 in adipocytes (Fig. 3f). The largest cluster of proteins induced by IF in a p53-dependent manner (cluster C3 in Fig. 3f) was enriched in proteins involved in inflammatory processes, namely IL-5 type 2 immunity, IL-2 signalling, and phagocytotic processes (Fig. 3g). This was further confirmed by an overall reduced mRNA expression of inflammatory markers in the HFD-IF-KO group measured in the adipocyte-rich fraction of eWAT (Supplementary Fig. 3g). As p53 suppression is known to impact cell cycling[32] (Supplementary Fig. 2i), we quantified staining with the proliferation marker Ki67 in HFD-IF and HFD-IF-KO sections. In CLS of eWAT, p53 KO led to a trend in reduction of Ki67-positive nuclei which was mirrored in the expression of proliferation markers in the adipocyte-rich eWAT fraction (Supplementary Fig. 3h–j). Furthermore, apoptotic proteins detected in the proteomics data were induced by IF in a p53-dependent manner (Supplementary Fig. 3k).

Together, our complementary omics analyses of eWAT in obese mice reveal a striking tissue remodelling and metabolic reprogramming upon an IF regimen, whereas eWAT seems refractory to most of these IF-mediated changes when p53 is depleted in adipocytes.

## Adipocyte-p53 coordinates adipocyte-immune cell interaction and the infiltration of lipid-associated macrophages in intermittently fasted mice

To further study the underlying p53- and IF-dependent mechanisms of adipocyte crosstalk with other AT-resident cell types, we analysed ligand-receptor interactions using our eWAT snRNA-seq data. The global interaction strength between adipocytes and immune cells was reduced in eWAT of HFD-IF-KO mice compared to HFD-IF (Fig. 4a), whereas the communication strength of adipocytes with other AT-resident cell types, like endothelial cells, remained unchanged. This data indicates that adipocytes directly signal to immune cells coordinating their infiltration and/or polarisation. The weakened interaction strength was reflected, for example, in reduced interactions of the Semaphorin/Plexin axis (Supplementary Fig. 4a, specifically Sema6D/Plexin1-Trem2 interaction), which was described previously as a regulator of AT remodelling[33]. However, this weakened interaction strength mainly corresponds to decreased receptor expression in the immune cell fraction in the HFD-IF-KO group (Supplementary Fig. 4b). On the other hand, in the HFD-IF-KO group, we noted that *Angptl4* expression was specifically decreased in the adipocyte fraction (Supplementary Fig. 4c) and the interaction with integrin receptors *Itga5* and *Itgb1* was lost (Supplementary Fig. 4a). In line with previous data showing that *Angptl4* is a fasting-induced, secreted factor[34], the expression of *Angptl4* was increased in differentiated C3H10T1/2 cells in response to starvation and quickly reduced to basal levels within the

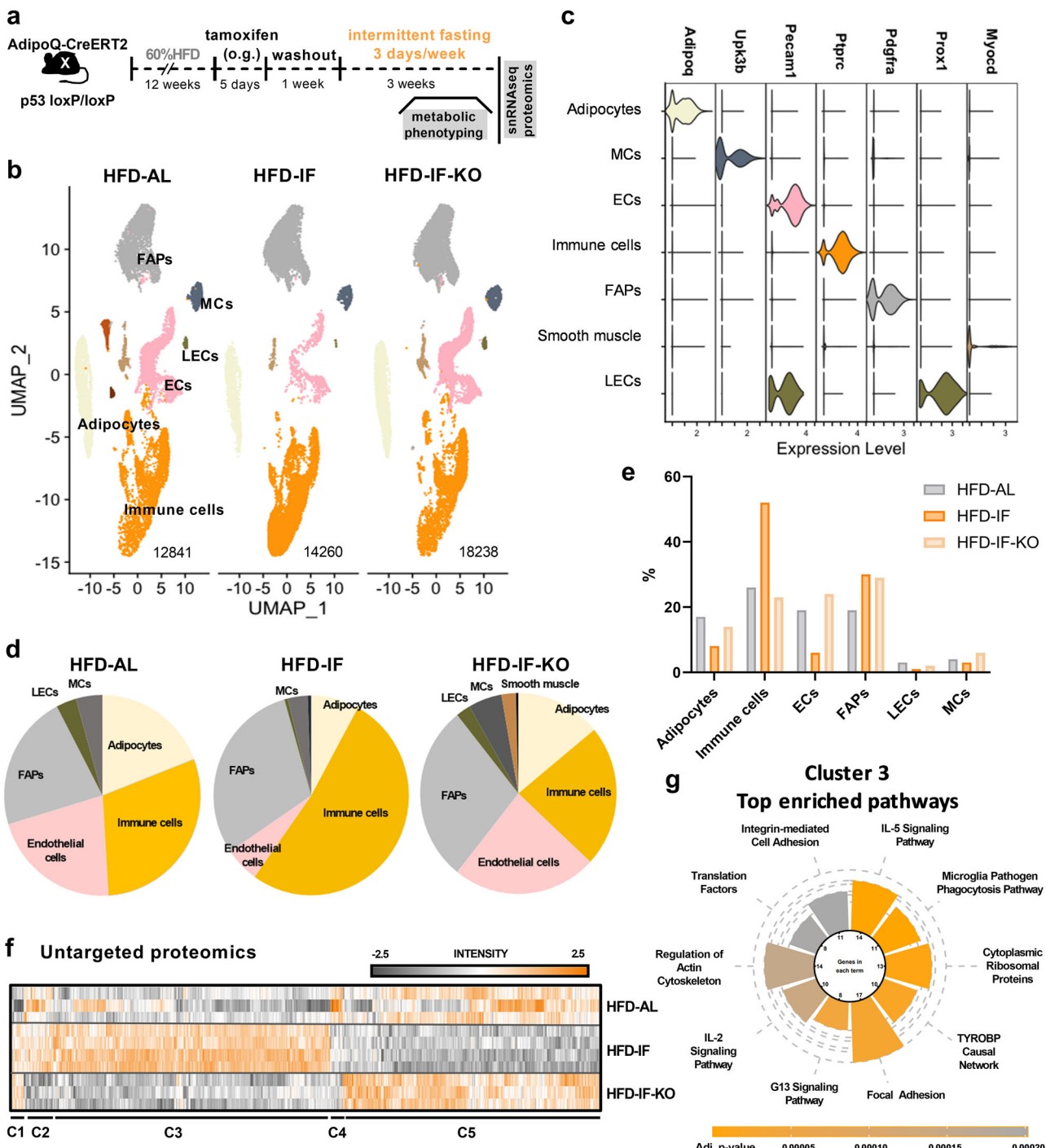

**Fig. 3 | Stress-induced p53 regulates eWAT remodelling in response to IF.**
**a** Experimental timeline in transgenic mice with adipocyte-specific, inducible p53 knock out (HFD-IF-KO). **b** UMAP plots derived from single-nuclei RNA sequencing with annotation of major clusters. Number of unique nuclei are indicated for each group. FAPs fibro-adipogenic progenitors, ECs endothelial cells, MCs mesothelial cells, LECs lymphatic endothelial cells. **c** Violin plots showing expression of one representative marker gene for each annotated subpopulation. **d**, **e** Pie charts (**d**) and bar graphs (**e**) showing the percentage of annotated cell types in eWAT of HFD-AL, HFD-IF, and HFD-IF-KO mice. **f** Heatmap of untargeted proteomics analysis from lysates derived from eWAT of HFD-AL ($n = 3$ mice), HFD-IF ($n = 4$ mice), and HFD-IF-KO ($n = 3$ mice) mice. Detected proteins of all samples were ANOVA tested (FDR 0.05) and submitted to hierarchical clustering using Perseus. Values are displayed as $z$-scores. **g** Pathway analysis using Wilcoxon Rank Sum test with Bonferroni multiple testing correction of cluster C3 (IF-induced, p53-dependent) of proteomic analysis. Source data are provided as a Source Data file.

first hours of refeeding (Supplementary Fig. 4d), reminiscent of the profile of p53 target genes (Supplementary Fig. 2h). Furthermore, we confirmed *Angptl4* downregulation after p53 knock-down in human differentiated adipocytes (Supplementary Fig. 4e), in line with an enriched binding of p53 to a binding site in the *Angptl4* gene (Supplementary Fig. 4f). It also exemplifies the importance of adipocyte

p53 signalling in the control of adipocyte-immune cell crosstalk, prompting us to further study the AT immune niche in response to IF and p53 status.

Therefore, to get a comprehensive map of all immune cells in eWAT, we divided the immune cell population into sub-clusters that revealed eight distinct cell populations, comprising T cells, B cells, and

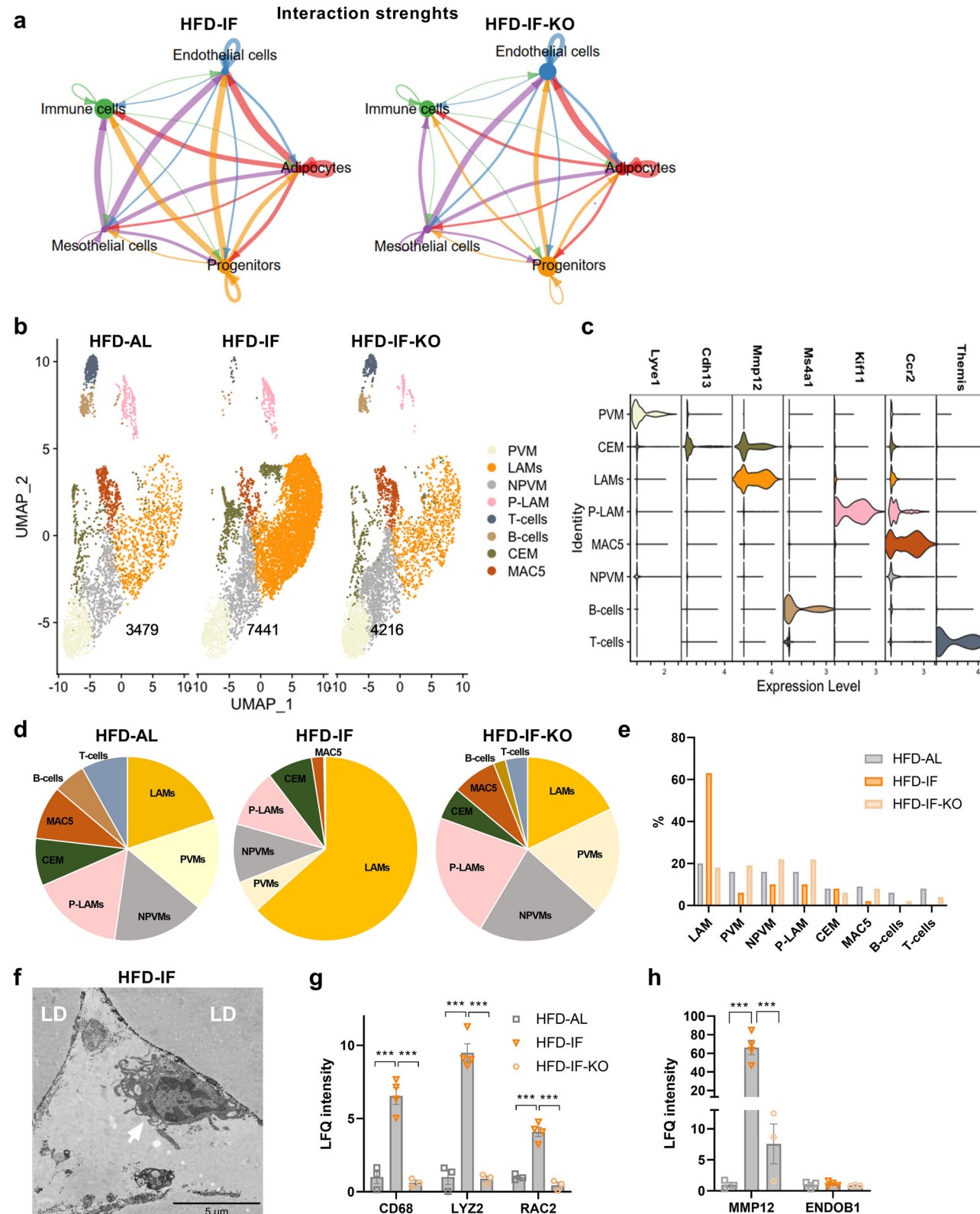

several macrophage subpopulations. The latter include LAMs, proliferating LAMs, perivascular macrophages (PVMs), non-PVMs, and collagen-producing macrophages (Fig. 4b), according to a recently published nomenclature[35] based on gene signatures (Fig. 4c). As previously described[2], macrophages were found as the predominant immune cell type in eWAT of obese mice (Fig. 4b, d, e). Importantly, LAMs mainly accounted for the inflammatory phenotype of eWAT in

the HFD-IF group, increasing from 20% of the immune fraction under ad libitum conditions to >60% upon IF (Fig. 4d, e). In contrast, other macrophage subtypes were either reduced or unchanged by IF (Fig. 4d, e). Furthermore, the fraction of T and B cells decreased in response to IF (Fig. 4b, d, e), emphasising that adaptive immune cell fractions in AT might also be influenced by nutritional cues. Importantly, most of the above mentioned IF-mediated alterations in the

**Fig. 4 | Lipid-associated macrophages strikingly increase in eWAT of inter-mittently fasted mice. a** Chord diagram showing the interaction network of different cell subpopulations in eWAT of HFD-IF and HFD-IF-KO mice, as determined by Cellchat (interaction strength is encoded in line thickness). **b** UMAP projections of annotated immune cell subpopulations (Lipid-associated macrophages (LAM), proliferating LAMs (P-LAM), perivascular macrophages (PVM), collagen-expressing macrophages (CEM), non-perivascular macrophages (NPVM), B cells, and T cells) in eWAT of HFD-AL, HFD-IF, and HFD-IF-KO mice. Number of unique nuclei are indicated for each group. **c** Violin plots showing expression of representative marker genes of each immune cell subpopulation. **d, e** Pie charts (**d**) and bar graphs (**e**) showing the relative percentage of each immune cell subpopulation in eWAT of HFD-AL, HFD-IF, and HFD-IF-KO mice. **f** Representative electron micrograph showing a lipid droplet-containing macrophage (arrow) in close proximity to an adipocyte (right, LD, unilocular lipid droplet) in eWAT of HFD-IF mice (representative out of 10 micrographs taken for each tissue from 3 mice per group). **g, h** LFQ intensity derived from the proteomics dataset of general macrophage markers (**g**), LAM-specific MMP12, and PVM-specific ENDOB1 (**h**) in eWAT of HFD-AL ($n = 3$ mice), HFD-IF ($n = 4$ mice), and HFD-IF-KO ($n = 3$ mice) groups. Data are presented as mean values ± SEM. Significant differences were analysed by one-way ANOVA with Bonferroni post hoc tests (**g, h**). ***$p < 0.001$. Exact $p$ values: **g** HFD-AL vs. HFD-IF: CD68: 0.0002, LYZ2: <0.0001, RAC2: 0.0002; HFD-IF vs. HFD-IF-KO: CD68: 0.0001, LYZ2: <0.0001, RAC2: <0.0001; **h** HFD-AL vs. HFD-IF: MMP12: 0.0003, HFD-IF vs. HFD-IF-KO: MMP12: 0.0005. Source data are provided as a Source Data file.

eWAT immune compartment were absent in the HFD-IF-KO group, underscoring a major role for p53 in AT inflammation upon IF.

Hence, we sought to further scrutinise the remarkable IF-mediated and p53-dependent increase in the abundance of LAMs. Using electron microscopy, we found macrophages in the IF group detected in close proximity to adipocytes, suggestive of them being LAMs (Fig. 4f). Using our proteomics data set, we confirmed an IF- and p53-mediated upregulation of proteins representing general macrophage markers (CD68, LYZ2 and RAC2 (Fig. 4g)). In line with snRNA-seq data, protein levels of the LAM-marker MMP12 were significantly upregulated by IF in an adipocyte p53-dependent manner, whereas expression of the PVM marker ENDOB1 remained unchanged between groups (Fig. 4h). Although TREM2 could not be detected by the proteomics approach, ablation of p53 in adipocytes caused a significant decrease in mRNA expression of *Trem2* in bulk eWAT and a trending reduction in sWAT, but not in BAT (Supplementary Fig. 4e–g). The effect of p53 KO on *Trem2* expression was also observed in eWAT of lean mice, whereas *Adgre1* (encoding the common macrophage marker F4/80) mRNA levels remained unchanged (Supplementary Fig. 4h).

These results show that the IF-mediated increase in the inflammatory phenotype of eWAT was mainly determined by an increase in LAM abundance, which was largely dependent on intact p53 signalling in adipocytes of intermittently fasted HFD-fed mice.

### Adipocyte p53 shapes the systemic response to IF
We next aimed to determine the systemic effects of adipocyte-specific p53 ablation in response to IF in HFD-fed mice. Interestingly, we observed increased body weight loss of p53 KO mice upon IF (Fig. 5a, b), concomitant with a reduction in sWAT weight and a trend towards reduced eWAT weight (Fig. 5c). We observed no overt changes in BAT and liver weight (Fig. 5c). While eWAT adipocyte size was unchanged, we observed a significant reduction in sWAT adipocyte size in HFD-IF-KO mice as compared to HFD-IF mice (Supplementary Fig. 5a–c). In line with reduced weight loss and adipocyte size, plasma non-esterified fatty acid (NEFA) levels were significantly higher than in HFD-IF mice (Fig. 5d). Plasma glycerol levels tended to increase (Supplementary Fig. 5d), whereas plasma triglyceride, cholesterol, and LDL concentrations remained unchanged (Supplementary Fig. 5e–g). In addition, a trend towards increased FA release from eWAT explants of HFD-IF-KO mice was observed, indicating that ablation of p53 in adipocytes increased their catabolic response (Fig. 5e). Furthermore, plasma ketone bodies were increased in HFD-IF-KO compared to HFD-IF mice (Fig. 5f), indicating improved fasting efficiency upon deletion of p53 in adipocytes. At the systemic level, insulin (ITT, Fig. 5g, h) and glucose (ipGTT, Supplementary Fig. 5h, i) tolerance tests revealed a substantial improvement in insulin sensitivity in HFD-IF-KO compared with HFD-IF mice, approaching levels of lean mice. This was accompanied by reduced fasting plasma insulin (Fig. 5i) and glucose levels (Supplementary Fig. 5j). Additionally, we detected a significant increase in insulin receptor (INSR) protein expression in the HFD-IF-KO mice compared to HFD-AL and HFD-IF groups (Fig. 5j),

suggesting that adipocyte-specific ablation of p53 upon IF renders mice more metabolically flexible. In support of this notion, indirect calorimetry revealed an increase in the respiratory exchange ratio (RER) of p53 KO mice during the refeeding phase (Fig. 5k), whereas food intake, locomotor activity, and energy expenditure remained comparable (Supplementary Fig. 5k–m). Furthermore, expression of the cytokine *Tnfα*, which causes insulin resistance[36], was significantly reduced in p53 KO mice (Fig. 5l), whereas the abundance of adipokines, such as adiponectin, leptin, and resistin, remained unaltered in plasma (Supplementary Fig. 5n).

In summary, p53 ablation in adipocytes after established obesity amplified the response to fasting and, thus, the long-term systemic health benefits of IF through improved metabolic flexibility.

### p53 ablation in adipocytes de-represses catabolic and oxidative gene expression.
As we observed exacerbated weight loss, increased plasma NEFAs, and heightened lipolysis in explants in the HFD-IF-KO group (Fig. 5d, e), we sought to further investigate adipocyte autonomous transcriptional signatures in the p53 KO group under IF. First, we generated a list of differentially expressed (DE) genes (Supplementary Data 1 and 2) in the major cell clusters from our snRNA-seq data set. Performing overrepresentation analysis with the gene set upregulated in the adipocyte cluster of the HFD-IF-KO compared to the HFD-IF group, yielded a strong enrichment in GO-biological processes terms related to lipid catabolism (Fig. 6a). This was confirmed by increased mRNA expression of lipolysis-related genes in adipocytes isolated from eWAT (Fig. 6b) and by augmented protein abundance of the main triglyceride hydrolase ATGL in eWAT of p53 KO mice (Fig. 6c). It should be noted that our measurements of mRNA and protein levels of catabolic enzymes might be confounded by the large number of lipid-laden macrophages in the adipocyte-rich fraction of eWAT and in the SVF of bulk AT. However, reinforcing a functional role of p53 in the regulation of catabolic genes in adipocytes, we detected a significant increase in *Lipe* (=*Hsl*), *Mgl*, and *Plin1* in bulk, non-inflamed eWAT from lean p53 KO mice (Fig. 6d), where macrophage marker *Adgre1* expression was unchanged (Fig. 6e). Consistent with an increased catabolic state of p53 KO adipocytes, adrenergic receptor signalling was an enriched GO-BP term in our DE analysis (Fig. 6a). In addition, transcript abundance of the major murine β-adrenergic receptor *Adrb3* was markedly induced in isolated eWAT adipocytes from HFD-IF-KO mice (Fig. 6f) and in nuclei classified as adipocytes in our snRNA-seq data (Fig. 6g). To investigate the metabolic fate of liberated fatty acids, we measured expression of fatty acid oxidation (FAO)-related genes in the adipocyte-rich fraction from eWAT. FAO genes were strongly increased in the HFD-IF-KO compared to the HFD-IF group (Fig. 6h). Further supporting p53-mediated FAO regulation, lipid-oxidative pathways were enriched in starved SGBS adipocytes with p53 knockdown (Supplementary Fig. 6a, b), and cluster 5 from our proteomics data (Fig. 3f), containing IF-downregulated proteins with higher expression in the HFD-IF-KO group mapped to pathways related to oxidative phosphorylation and TCA cycle (Fig. 6i).

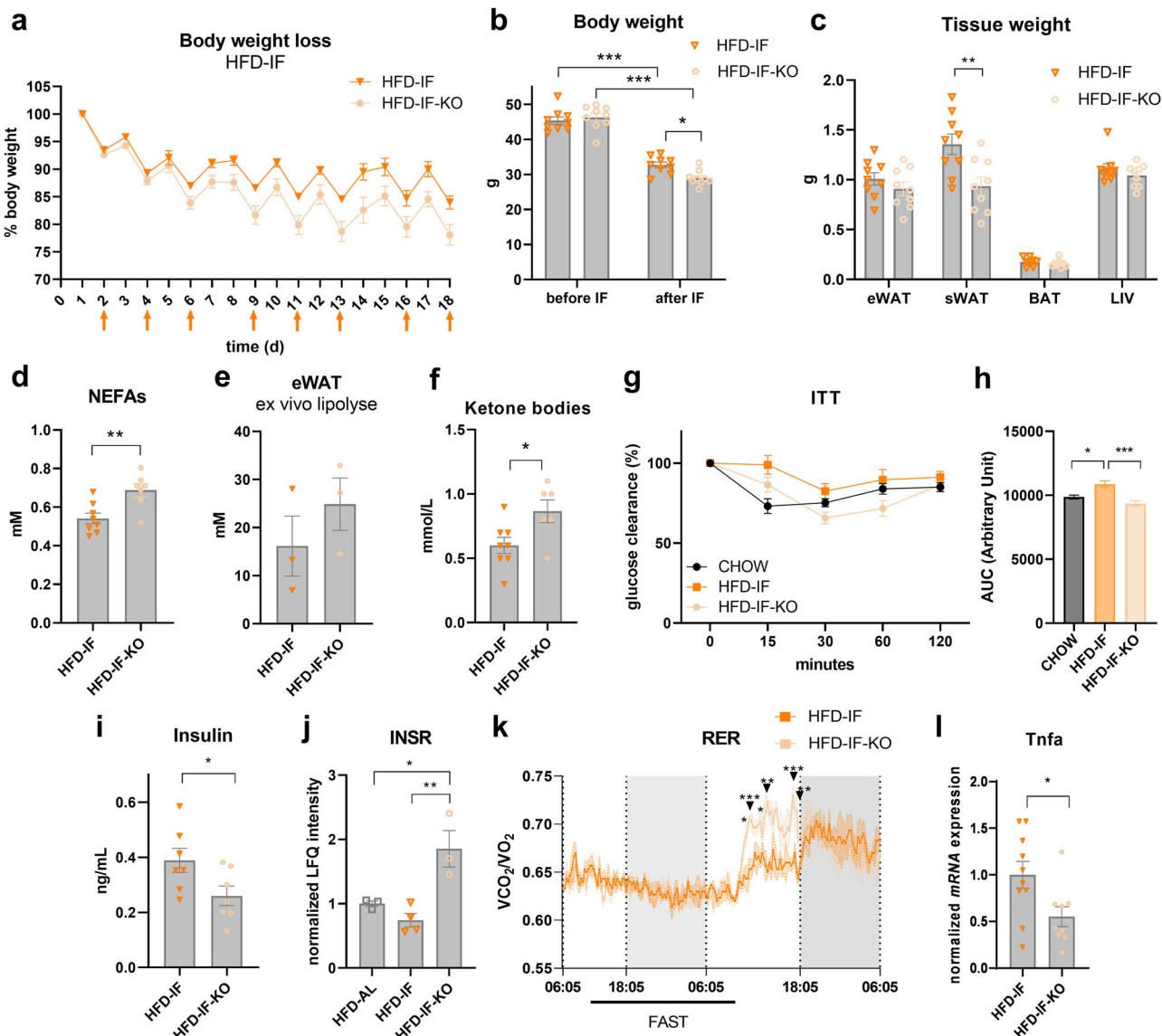

**Fig. 5 | Adipocyte p53 shapes the systemic response to IF. a** Body weight loss of HFD-IF ($n = 3$ mice) and HFD-IF-KO ($n = 6$ mice) mice. **b** Body weight before and after IF of HFD-IF and HFD-IF-KO ($n = 9$ mice per group) mice. **c** Adipose depot and liver (LIV) weight of HFD-IF and HFD-IF-KO ($n = 9$ mice per group) mice. **d** Plasma NEFA levels of HFD-IF ($n = 8$ mice) and HFD-IF-KO ($n = 7$ mice) mice. **e** Ex vivo lipolysis assay showing fatty acid release (mM) per hour (h) normalised to protein concentration of eWAT explants from HFD-IF or HFD-IF-KO mice ($n = 3$, fat pads from 3 mice). **f** Plasma ketone bodies of HFD-IF ($n = 8$ mice) and HFD-IF-KO ($n = 6$ mice) groups. **g, h** Insulin tolerance test (ITT) of chow-fed ($n = 8$ mice), HFD-IF ($n = 10$ mice), and HFD-IF-KO ($n = 11$ mice) groups. **i** Plasma insulin levels of HFD-IF and HFD-IF-KO ($n = 7$ mice per group) groups. **j** Normalised proteomics LFQ intensity of INSR in eWAT of HFD-AL ($n = 3$ mice), HFD-IF ($n = 4$ mice), and HFD-IF-KO ($n = 3$ mice) mice. **k** Respiratory exchange ratio (RER) of HFD-IF and HFD-IF-KO

($n = 3$ mice per group) groups during a time-period of 48 h analysed by indirect calorimetry. **l** *Tnfa* expression analysed in HFD-IF ($n = 10$ mice) and HFD-IF-KO ($n = 9$ mice) groups. Data are presented as mean values ± SEM. Significant differences were analysed by two-tailed, unpaired *t*-test (**c**–**e**, **f**, **i**, **l**) or by one-way (**h**, **j**) or two-way (**a**, **b**, **g**, **k**) ANOVA with Bonferroni post hoc tests. ***$p < 0.001$, **$p < 0.01$, and *$p < 0.05$. Exact *p* values: **b** HFD-IF before vs. after: <0.0001, HFD-IF-KO before vs. after: <0.0001, after IF HFD-IF vs. HFD-IF-KO: 0.0490; **c** HFD-IF vs. HFD-IF-KO sWAT: 0.0001; **d** 0.0049; **f** 0.0259; **h** CHOW vs. HFD-IF: 0.0166; HFD-IF vs. HFD-IF-KO: <0.0001; **i** 0.0414; **j** HFD-AL vs. HFD-IF: 0.0301, HFD-IF vs. HFD-IF-KO: 0.0055; **k** 13:36: $p = 0.0325$, 13:51: $p = 0.0002$, 14:21: $p = 0.0133$, 15:0:6 $p = 0.0052$, 17:21: $p = 0.0007$, 17:51 $p = 0.0052$; **l** 0.0241. Source data are provided as a Source Data file.

Collectively, several lines of evidence suggest that p53 acts as a catabolic break upon IF, which is released through adipocyte-specific ablation of p53 signalling in obese mice.

### p53 is associated with weight loss retention in obese humans
To interrogate the translational relevance of our findings, we stratified diabetic patients undergoing three cycles of a fasting-mimicking diet (FMD[37]) according to a metabolically relevant single nucleotide polymorphism (SNP) in p53 at amino acid 72 (rs1042522), encoding for either an arginine (R72) or a proline (P72)[38,39]. In this German FMD

cohort of 20 patients, 14 were homozygous for the R72 variant and six carried the P72 variant. *TP53* expression levels in isolated white blood cells were similar in carriers of the two variants (Supplementary Fig. 7a). The R72/P72 frequencies correspond to the expected distribution in individuals living in the northern hemisphere[38]. After the FMD intervention, all patients had significantly decreased BMI (Supplementary Fig. 7b) and HbA1c levels (Supplementary Fig. 7c), proving the efficacy of FMD. In line with our data in mice, plasma IL-6 levels (Fig. 7a) were significantly higher after the FMD intervention, suggesting that the inflammatory phenotype after dietary interventions

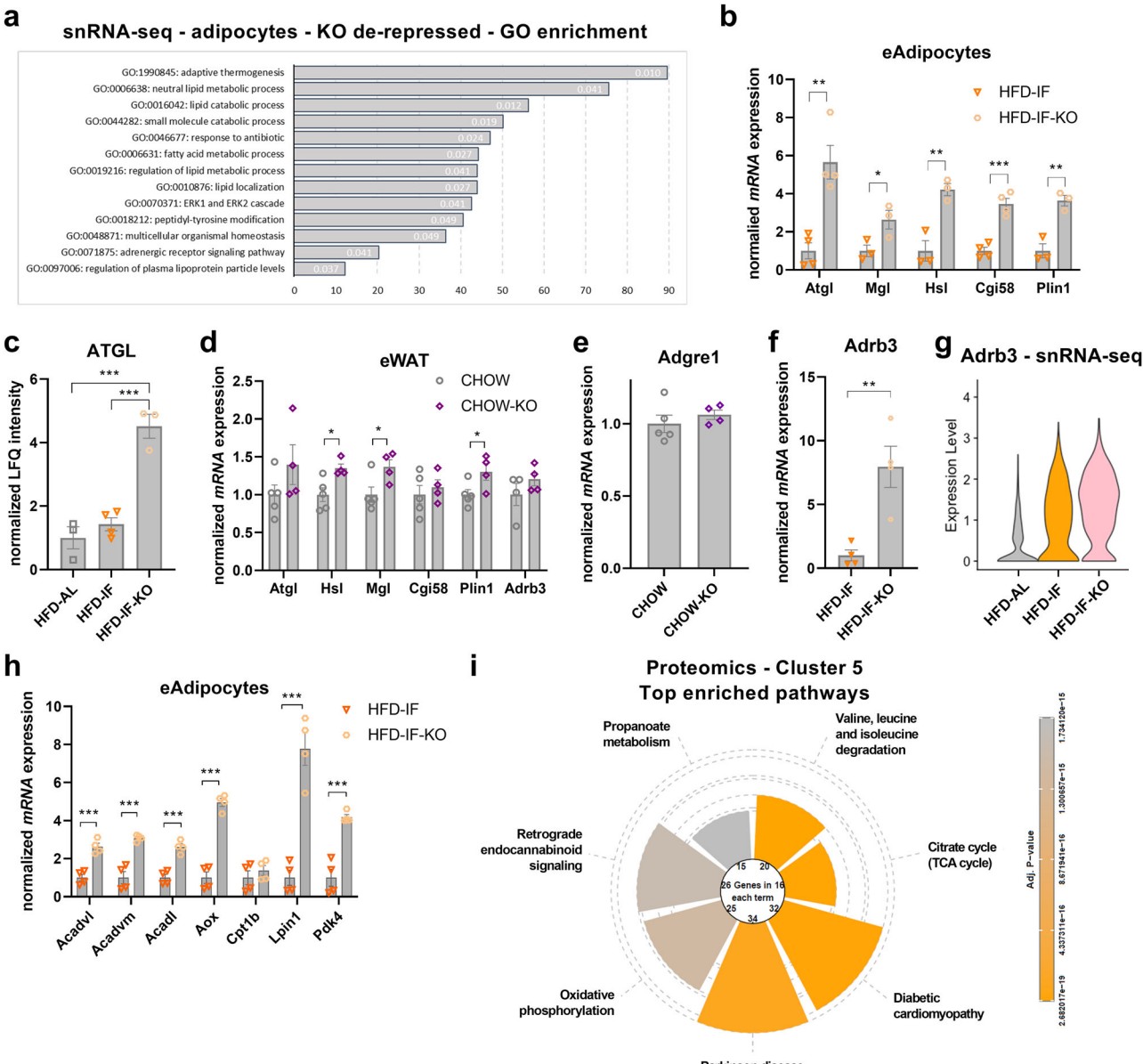

**Fig. 6 | p53 ablation in adipocytes de-represses catabolic and oxidative gene expression programmes. a** Differentially expressed genes between HFD-IF-KO and HFD-IF groups were extracted from the single nuclei RNA-seq data set (Supplementary Data 2). Genes de-repressed by p53 KO in the adipocyte cluster were mapped to gene ontology (GO) biological processes. Categories with FDR < 0.05 (FDR value given in bars) are shown, with enrichment ratios on the *x*-axis. **b** mRNA expression levels of lipolysis-associated genes in the adipocyte-rich fraction isolated from eWAT of HFD-IF and HFD-IF-KO ($n = 3–4$ mice per group) mice. **c** Normalised proteomics LFQ intensity of ATGL of HFD-AL ($n = 3$ mice), HFD-IF ($n = 4$ mice), and HFD-IF-KO ($n = 3$ mice) groups. **d, e** mRNA expression levels of lipolysis-associated genes (**d**) and *Adgre1* (=F4/80) (**e**) in bulk eWAT of lean (CHOW, $n = 5$ mice) and p53 KO (CHOW-KO, $n = 4$ mice) mice. **f** *Adrb3* mRNA expression levels in the adipocyte-rich fraction isolated of eWAT of HFD-IF or HFD-IF-KO ($n = 4$ mice per group) mice. **g** Transcript abundance of *Adrb3* in adipocytes of HFD-AL,

HFD-IF, and HFD-IF-KO mice as determined by single-nuclei RNA sequencing. **h** mRNA expression levels of fatty acid oxidation-related genes in the adipocyte-rich fraction isolated of eWAT from HFD-IF and HFD-IF-KO ($n = 4$ mice per group) mice. **i** Pathway analysis using Wilcoxon Rank Sum test with Bonferroni multiple testing correction of cluster 5 (IF-repressed, elevated in HFD-IF-KO over HFD-IF) of proteomic analysis (Fig. 3f). Data are presented as mean values ± SEM. Significant differences were analysed by two-tailed, unpaired *t*-test (**b, d–f, h**) or by one-way ANOVA with Bonferroni post hoc tests (**c**). ***$p < 0.001$, **$p < 0.01$, and *$p < 0.05$. Exact *p* values: **b** *Atgl*: 0.00395, *Mgl*: 0.0465, *Hsl*: 0.0067, *Cgi58*: 0.0004, *Plin1*: 0.0047; **c** HFD-AL vs. HFD-IF-KO: 0.0003, HFD-IF vs. HFD-IF-KO: 0.0005; **d** *Hsl*: 0.0171, *Mgl*: 0.0365, *Plin1*: 0.0454; **f** 0.0061; **h**: *Acadvl*: 0.0008, *Acadvm*: 0.0007, *Acadl*: 0.0005, *Aox*: <0.0001, *Lpin1*: 0.0004, *Pdk4*: 0.0005. Source data are provided as a Source Data file.

may be conserved in humans, which is also consistent with a recent study in humans undergoing a single 10-day fasting bout[40]. After three cycles of FMD, patients carrying the P72 variant showed slightly reduced fat mass loss (Fig. 7b) and increased plasma triglycerides (Fig. 7c) and HbA1c (Fig. 7d), indicating a lower response to FMD in comparison to patients carrying the R72 variant. This reduced effect of FMD in the P72 over the R72 group was still evident in a refeeding phase, where patients had no nutritional restrictions for 1 week after

the third FMD cycle. This was reflected in increased average BMI regain (Fig. 7e), plasma triglycerides (Fig. 7f), HbA1c (Fig. 7g), and fasting plasma glucose (Fig. 7h) in the P72 group, indicating that this p53 polymorphism may affect both the immediate fasting response and long-term metabolic parameters after cessation of FMD, i.e., during the weight regain period. To estimate the effect of the P72R variant on inflammatory signalling, we overexpressed the P72 or R72 variant of *TP53* in p53 KO C3H10T1/2 adipocytes, before challenging the cells

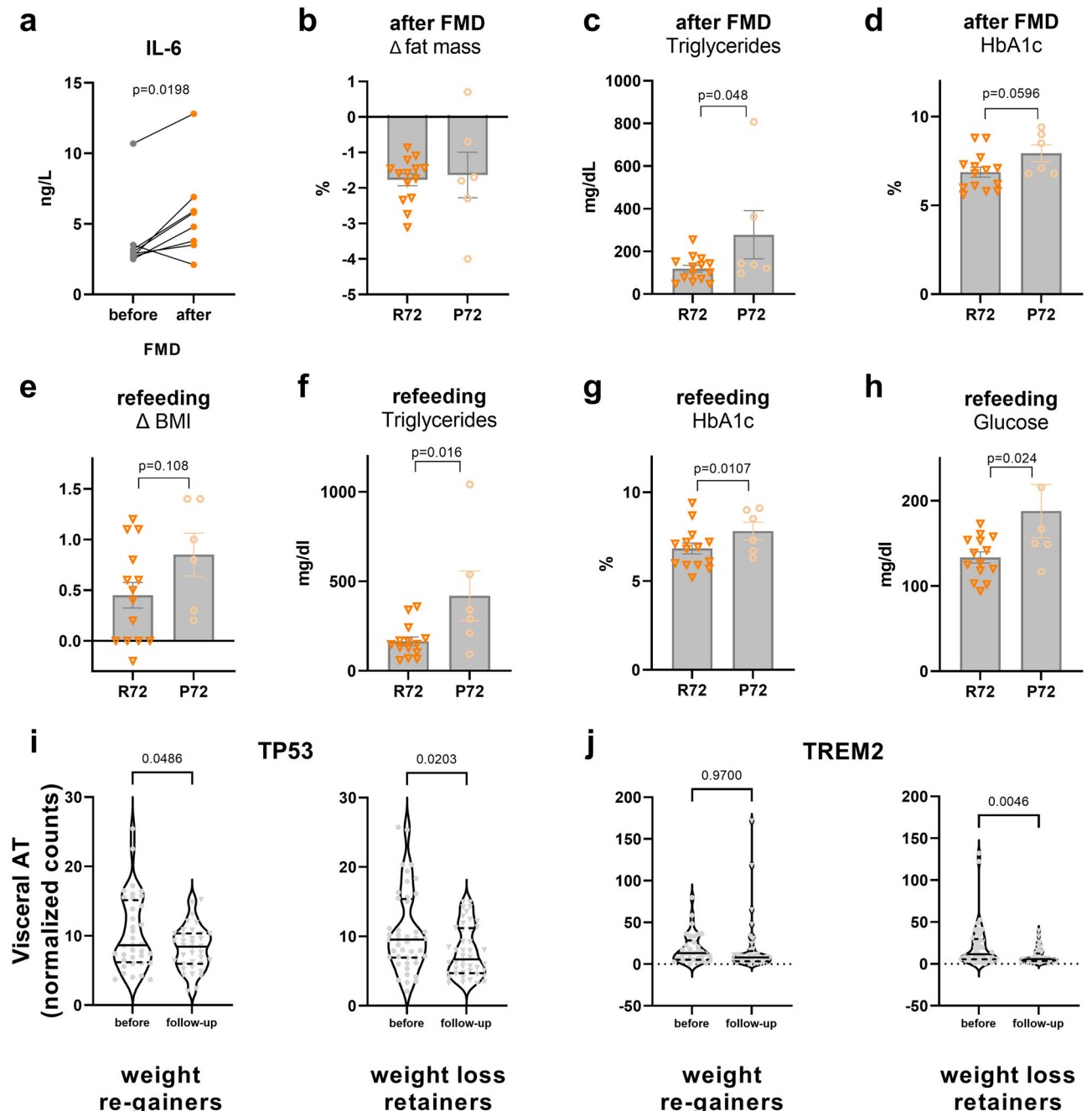

**Fig. 7 | p53 is associated with weight loss retention in obese humans. a** Plasma IL-6 levels before and after three cycles of fasting-mimicking diet (FMD) in a cohort of diabetic patients (*n* = 8). **b–d** Fat mass loss (in percent from baseline), plasma triglycerides, and HbA1c after FMD and stratification according to the *TP53* P72R polymorphism (R72, *n* = 14 patients; P72, *n* = 6 patients). **e–h** Body mass index (BMI) regain, plasma triglycerides, HbA1c, and glucose in the same cohort (R72, *n* = 14 patients; P72, *n* = 6 patients) after 1 week of refeeding after three FMD cycles.

**i, j** RNA-seq expression of *TP53* (**i**) and the LAM marker *TREM2* (**j**) in visceral AT from obese subjects that were collected at the time they underwent bariatric surgery and after follow-up of 2 years. Comparison between weight loss retainers (BMI ≥ 25% loss retained, *n* = 44 patients) and weight re-gainers (BMI < 25% loss retained, *n* = 36 patients). Data are presented as mean values ± SEM. Significant differences were analysed by two-tailed, paired (**a**, **i**, **j**) or unpaired (**b–h**) *t*-test. Exact *p* values are given in figures. Source data are provided as a Source Data file.

with 24 h of starvation (Supplementary Fig. 7d). In this experimental setup, we assessed expression of *Ccl2*, a gene encoding for a cytokine that is involved in monocyte chemotaxis and that was formerly characterised as p53 target gene[41]. Overexpression of the R72 variant resulted in a significant reduction in *Ccl2* expression compared to overexpression of the P72 variant (Supplemental Fig. 7e), suggesting blunted monocyte recruitment signalling consistent with improved fasting response in patients carrying the R72 variant (Fig. 7b–h).

To gain insight into surgery-induced weight loss followed by weight regain, we investigated AT biopsies from a cohort of obese

patients who underwent bariatric surgery 2 years prior to sampling. The cohort was stratified by individual weight loss efficiency (using a 25% BMI reduction to demarcate efficient and less efficient long-term weight loss retention). In visceral AT, reduction of *TP53* expression in the follow-up group, compared to levels before surgery, was stronger in weight loss retainers (>25% BMI reduction) than in weigh re-gainers, (Fig. 7i). Significantly lower expression in the weight loss retainer group was also found for *TREM2* (Fig. 7j) in visceral AT biopsies, while *TP53* and *TREM2* expression changes were less pronounced in subcutaneous samples (Supplementary Fig. 7f, g).

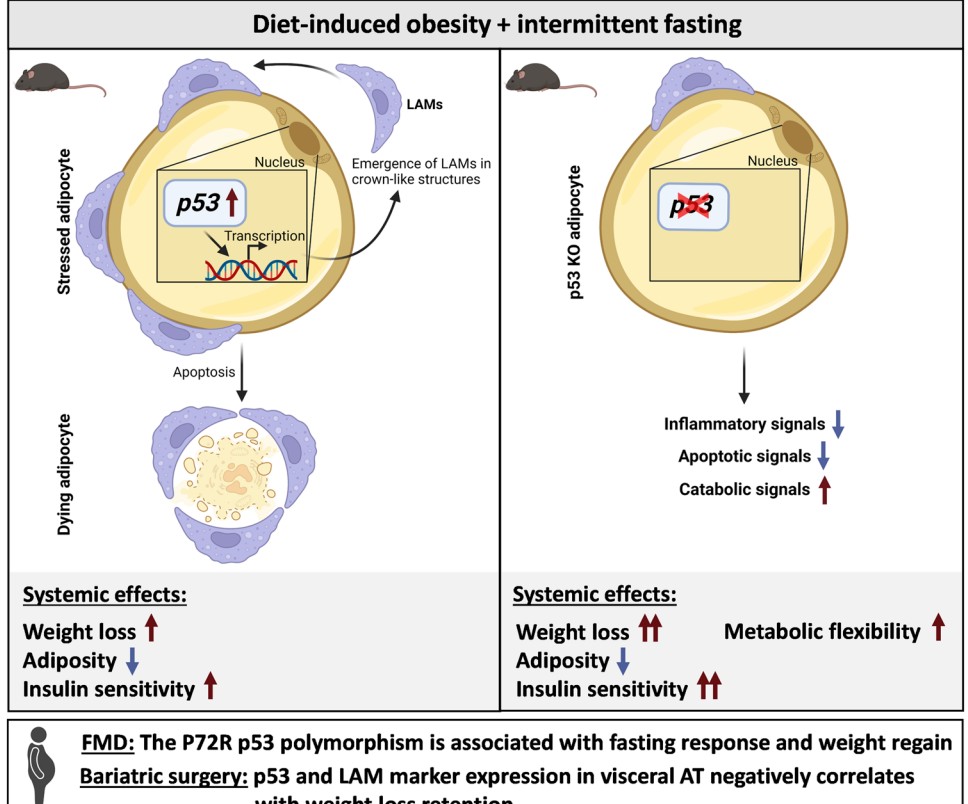

**Fig. 8 | Scheme of key findings.** In the context of obesity, intermittent fasting increases the abundance of lipid-associated macrophages (LAMs) in crown-like structures of visceral adipose tissue of mice. This coincides with adipocyte apoptosis and activation of p53 signalling. Knock out (KO) of p53 specifically in adipocytes reduces inflammatory and apoptotic signalling, while elevating the catabolic state of adipocytes. Consequently, p53 KO leads to increased weight loss, enhancing the metabolic health benefits of intermittent fasting. Data in human cohorts implicate p53 in clinical weight loss scenarios. Created with BioRender (www.biorender.com).

Together with our findings in mice, these data suggest that p53 signalling, and possibly p53-dependent LAM recruitment to AT, may limit the effectiveness of weight loss and weight loss retention.

## Discussion

AT is a highly heterogeneous and plastic tissue consisting of various cellular components. Dissecting WAT on single-cell or single-nuclei level has been instrumental for understanding its complexity in the context of obesity[35,42], weight-cycling[9], cold exposure[43], and exercise[44], as comprehensively reviewed recently[20]. Here we provide a complete cellular map of changes in AT composition in response to IF and p53 status in obese mice. We found that IF caused extensive, adipocyte p53-dependent tissue remodelling, with emergence of LAMs as one key feature. Whereas the abundance of other cell types in AT (like e.g. endothelial cells) was also strongly affected by IF, data derived from ligand-receptor analysis and in vitro experiments indicated a prominent, direct p53-dependent signalling only between adipocytes and immune cells. Adipocyte-specific loss of p53 augmented the metabolic health benefits of IF, thus representing a novel mechanism that could be therapeutically utilised to boost the effectiveness of dietary interventions and to overcome retained AT dysfunction upon weight loss in formerly obese patients. Thus, our findings (summarised in Fig. 8) have multiple implications on our understanding of the physiology of cyclic fasting. (1) IF induces p53-driven stress response pathways in adipocytes, subsequently altering the transcriptional profile of adipocytes; (2) p53-mediated cell death may be responsible for a certain degree of visceral adipose depot shrinkage upon weight loss; (3) our data suggest a model that elevated fat cell death upon IF attracts LAMs and/or induces the proliferation of LAMs in AT, worsening the inflammatory state of AT; (4) adipocyte-specific depletion of p53 impedes excessive macrophage abundance under IF and (5) improves the systemic response to fasting, as evident by an increased catabolic state of adipocytes, accelerated body weight loss, improved systemic insulin sensitivity, and enhanced metabolic flexibility of adipocytes; furthermore, (6) our findings in mice may be translatable to humans, as the metabolic response of diabetic patients undergoing FMD varied as a function of p53 status, and the efficiency of weight loss retention of individuals after bariatric surgery correlated negatively with *TREM2* and *TP53* expression levels in visceral AT.

Previous studies showed that weight loss induced by dietary and/or lifestyle interventions retained or even worsened AT inflammation[6–16]. Furthermore, pharmacological induction of catecholamine signalling or acute fasting was shown to trigger the formation of crown-like structures[45]. In addition to an overall increase in the number of macrophages, nutrient deprivation may affect macrophage polarisation and/or the abundance of specific macrophage subtypes[46–48]. Cyclic IF in obese mice increased the number of anti-inflammatory M2 macrophages, without affecting the abundance of M1 macrophages[48]. Moreover, another study confirmed an increase in M2 versus M1-like macrophages during nutrient deprivation, although this process was impaired in obese mice[47]. However, the creation of single cell methods-based maps of the immune niche in AT highlighted that the M1/M2 paradigm is oversimplified and that macrophages can polarise to a broad spectrum of different states. Single-cell RNA-seq on the SVF of healthy mice demonstrated that perivascular macrophages are rapidly replenished upon a single fasting bout or β-adrenergic activation[46], underscoring that different macrophage subtypes flexibly respond to catabolic stimuli. Importantly, the obesity-induced

inflammatory landscape of AT was shown to be imprinted during weight loss and even worsened during weight regain, which was accompanied by an increase in LAMs in weight-cycling mice[9]. In line, HFD-induced alterations in the chromatin landscape of ATMs were conserved after weight loss, which may render ATMs of formerly obese mice susceptible to angiogenic and pro-inflammatory stimuli[49]. These data are in accordance with studies in humans showing that the number of inflammatory hotspots in visceral AT upon weight reduction was highly variable and predictive for weight maintenance[50]. Together, weight loss, especially in obesity, elicits AT remodelling that might limit the effectiveness of weight loss interventions. However, weight loss-induced AT inflammation could be a temporary process to restore tissue homoeostasis, and the inflammatory phenotype might be resolved upon prolonged lifestyle interventions or persistent weight maintenance[12,13,51].

In addition to an increased number of macrophages in response to weight loss, our data after IF show that they undergo a transcriptional and compositional shift towards LAMs that have been reported to have heightened lipid catabolism[29]. LAMs represent a distinct class of crowning macrophages that surround adipocytes prone to cell death to clear cellular remnants and metabolise liberated lipids[52]. LAMs differ from other macrophage subtypes by their high lysosomal and phagocytotic activity, thereby preserving tissue integrity under metabolically demanding conditions[29]. The strong phagocytotic capacities of LAMs might explain the physiological mechanism of LAM formation upon IF. Following this logic, clearance of lipid-rich debris derived from dying adipocytes by LAMs could prevent toxic effects on neighbouring cells. Data on the function of LAMs in limiting the obesity phenotype have been contradictory, as genetic deletion of *Trem2* improved metabolic health in obese mice[29,53,54], but these effects were not related to the function of *Trem2* in immune cells[53]. In the liver, Trem2-positive macrophages were shown to be responsible for the clearance of apoptotic hepatocytes, which prevented non-alcoholic steatohepatitis development[55,56]. Therefore, ablation of p53 in adipocytes might disrupt the cycle of fat cell apoptosis and LAM recruitment. From a translational perspective, a recent study found a positive correlation of *TREM2* expression levels in AT with BMI only in males and between a specific TREM2 polymorphism and BMI in female individuals[57]. An association of LAM abundance in visceral AT and the degree of obesity was confirmed by another study[29]. Another study in humans reported a positive correlation between circulating TREM2 concentrations and insulin resistance, although there was no association with body weight[58]. These data are in agreement with the well-described association of macrophage-dependent chronic AT inflammation with systemic insulin resistance and with genetic knock out of key inflammatory signalling molecules, scaffolding proteins, or cytokines sufficient to dissociate obesity from insulin resistance[59–66]. However, anti-inflammatory therapies so far failed to consistently ameliorate obesity-associated diseases in humans[67]. Thus, it is essential to define adipocyte-intrinsic transcriptional regulators that recruit immune cells to AT to find novel druggable targets.

In this context, we found that the induction of the transcription factor p53 in adipocytes is required for the regulation of AT LAM abundance in response to IF. p53 has been extensively studied in the context of cancer development, as p53 is the most frequently mutated gene in human cancer and defined as the gatekeeper of cellular fate[68]. However, more recent studies identified important roles of p53 in non-cancerous tissue homoeostasis by inducing canonical downstream pathways like apoptosis, senescence, and DNA damage response[69]. In relation to AT remodelling, it was shown that p53 signalling is induced in adipocytes upon obesity development in mice, causing adipocyte senescence, AT inflammation, and systemic insulin resistance[24,70]. Here, we found that IF promotes adipocyte stress by activating apoptosis as a canonical downstream signalling pathway of p53[71]. In cultured adipocytes, we also show that p53 signalling activation by

starvation is reversible by refeeding and that it elicits not only an apoptotic but also an inflammatory gene programme. Generally, p53 is considered as a stress responsive protein that activates different downstream pathways depending on the magnitude, duration, and type of upstream stressors. Previous studies reported a number of nutrient-dependent upstream regulators of p53 in different (non)-cancerous cells, including AMPK, fatty acids, reactive oxygen species, and growth hormones[23]. Especially, AMPK as an important nutrient sensor[72] could act as mediator between energy shortage and p53 activation. Our prior work in liver demonstrated that p53 activation in response to fasting is dependent on AMPK[25]. Thus, we could surmise that fasting-activated AMPK signalling in adipocytes further aggravates the already heightened p53 signalling under HFD[70]. As a possible scenario leading to adipocyte death, the cycling between fasting and refeeding could constitute cellular stress from repeated shrinkage and enlargement of adipocytes causing mechanical stress against the less flexible extra-cellular matrix[18]. Adipocyte death is one of the main drivers for AT immune cell recruitment, as stress signals released by dying adipocytes trigger inflammatory programmes that stimulate cytokine release and the infiltration of macrophages[73–77]. Thus, p53-induced apoptosis in response to (cyclic) fasting may cause recruitment and/or proliferation of LAMs. Improved systemic metabolism under IF despite increased AT inflammation and adipocyte cell death, can be explained by IF-mediated effects on other organs important for metabolic homoeostasis. For example, in our model we show reduction of hepatic steatosis under IF. Furthermore, our ligand-receptor analysis indicates that the inflammatory phenotype is directly mediated via cellular interactions between adipocytes and immune cells, as we found a decrease in the interaction strength in p53 KO mice. However, further experiments are needed to understand how adipocytes cope with nutritional stress in the absence of p53 and if transcription-independent cell death programmes are utilised. Together, our data suggest that p53 controls a stress-responsive gene programme, rather than a single downstream gene, that facilitates fasting-dependent recruitment of macrophages into eWAT under metabolic stress.

At the systemic level, adipocyte-specific deletion of p53 augmented the response to IF, which was accompanied by a de-repressed lipid catabolic and oxidative state of adipocytes upon p53 KO in comparison to controls. Under healthy conditions, WAT mass loss is facilitated by lipid release in response to catabolic stimuli[78]. Beyond the p53 KO-mediated adipocyte-autonomous shift towards a catabolic gene programme, ATMs can also affect adipocyte lipid release by inducing catecholamine resistance[3], which is defined as a reduced catabolic response to physiological levels of catecholamines and can be caused by lowered expression of β-adrenergic receptors[79]. Thus, we hypothesise that visceral adipocytes in obese mice, maybe partly through the crosstalk with LAMs and partly by p53-driven transcriptional reprogramming, possess mechanisms to restrain catabolic stimuli upon IF. Deletion of p53 in adipocytes may attenuate this catabolic dampening, thereby (re-)sensitising adipocytes to these stimuli as indicated by increased expression of lipolytic mediators, the insulin receptor *Insr*, and the β-adrenergic receptor *Adrb3*, as well as by heightened lipid release from adipocytes. As a possible consequence, p53 KO mice showed an increased metabolic flexibility, indicated by improved systemic insulin sensitivity and heightened RER during the refeeding period. Thus, the p53 signalling pathway in adipocytes might represent an interesting target to ameliorate the effectiveness of dietary weight loss interventions.

Underscoring the therapeutic utility of p53 in obesity interventions, our data propose a translational relevance of p53 in the response to weight-reducing interventions in humans. Stratifying diabetic patients undergoing an FMD protocol according to the P72R variant showed that patients carrying the R72 variant had an improved fasting response that persisted after fasting cessation. P72 is the ancestral

variant and more common in populations living near the equator. The R72 variant is found primarily at higher geographical latitudes and has been described as metabolically disadvantageous, as it positively correlates with BMI and the susceptibility to diabetes[38]. Accordingly, "humanised" knock-in mice that carry the R72 variant showed increased body weight, adipose inflammation, and insulin resistance, compared to controls carrying the P72 variant[39]. Furthermore, the R72 variant was shown to harbour increased AMPK-sensitivity as measured by p53-Ser15 phosphorylation in glucose-deprived cancer cell lines and in liver and pancreas of mice treated with the AMPK activator AICAR[39]. This was accompanied by increased growth arrest and decreased apoptosis in nutrient-deprived cell lines. Therefore, the human R72 variant might preferentially induce senescence rather than cell death after nutrient deprivation[39]. This data suggests that the variants induce a different downstream transcriptional programme in response to metabolic stress, in line with our observation of a significantly reduced expression of the cytokine *Ccl2* in starved C3H10T1/2 adipocytes when we overexpressed the R72 compared to the P72 variant. Furthermore, in a cohort of obese individuals who had undergone bariatric surgery, expression of *TREM2*, and *TP53* in visceral AT correlated negatively with effective long-term weight loss retention.

Taken together, our data provide evidence that adipocyte p53 is a central determinant of macrophage abundance under IF in obese mice and of the effectiveness of the response to IF. They further suggest a connection of IF-mediated, p53-regulated cell death with LAM recruitment and the catabolic state of visceral AT, ultimately coordinating systemic insulin sensitivity and metabolic flexibility upon IF. Our study thereby suggests that failure to resolve the AT LAM niche after cyclic fasting may dictate an inflammatory obesogenic memory that pushes the organism towards metabolic inflexibility. Future studies should investigate whether p53 ablation in adipocytes, with concomitant reduction in the abundance of AT macrophages, might lead to differences in weight (re-)gain after return to normal or obesogenic diet, to further test the hypothesis that adipocyte p53 is key in establishing an immuno-obesogenic memory of AT that promotes weight regain.

## Methods

### Ethical regulations

All animal studies were approved by the Austrian Ministry for Education, Science and Research (Vienna, Austria, BMWFW-66.010/0087-WF/V/3b/2017) and performed strictly according to its guidelines. Human studies were performed in agreement with the Declaration of Helsinki, all patients provided written informed consent, and ethical approvals were granted from the University Hospital of Heidelberg (Ethic-Nr. S-682/2016) or by the Ethics Committee of the University of Leipzig (approval number 159-12-21052012).

### Cell lines

All cells were cultivated in a humidified atmosphere of 5% $CO_2$ and 95% air at 37 °C. C3H10T1/2 clone 8 mouse mesenchymal stem cells were purchased from ATCC (CCL-226, ATCC, Virginia, United States) and cultivated and differentiated as described previously[80]. Briefly, cells were maintained in growth medium (High-glucose Dulbecco's modified Eagle's medium (DMEM) supplemented with 10% FBS, 2 mM L-glutamine, 100 U/ml penicillin, 100 mg/ml streptomycin (all from Thermo Fisher Scientific Waltham, MA, United States)). Two days post-confluent cells were induced to undergo adipogenesis by addition of 1 µM dexamethasone (DEX), 500 µM 3-isobutyl-1-methylxanthine (IBMX), 5 µg/ml insulin, and 1 µM rosiglitazone (all Sigma St. Louis, MI, United States). Human Simpson-Golabi-Behmel syndrome (SGBS) preadipocyte cells were kindly provided by Martin Wabitsch and maintained and differentiated as described before[81,82]. Briefly, cells were cultured in DMEM/F12 supplemented with 10% FBS, 100 U/ml penicillin, 100 mg/ml streptomycin, 8 µg/ml biotin, and 4 µg/ml

pantothenic acid (all from Thermo Fisher Scientific Waltham, MA, United States). To induce differentiation, cells were grown to confluence and subsequently cultured in medium as above, but without FBS and containing 0.01 mg/ml transferrin, 1 µM cortisol, 200 pM triiodothyronine (T3), 20 nM human insulin, 0.25 µM DEX, 500 µM IBMX, and rosiglitazone (2 µM until day 4, 1 µM until day 8, then without rosiglitazone) (all Sigma St. Louis, MI, United States). Human multipotent adipose-derived stem cells (hMADs) were cultivated and differentiated to white adipocytes as previously described[83]. Briefly, cells were kept in growth medium consisting of DMEM with 10% FBS, 10 mM HEPES, 2 mM L-glutamine, 100 U/ml penicillin, 100 mg/ml streptomycin (all from Thermo Fisher Scientific Waltham, MA, United States), and 2.5 ng/ml hFGF-2 (all Sigma St. Louis, MI, United States). For adipocyte differentiation, cells were grown to confluence and medium was changed to growth medium without hFGF2. After 2 days, adipocyte differentiation was initiated with growth medium without hFGF2 and supplemented with 860 nM insulin, 10 µg/ml apo-transferrin, 0.2 nM T3, 100 nM rosiglitazone, 100 µM IBMX, and 1 µM DEX (all Sigma St. Louis, MI, United States). IBMX and DEX were omitted after 3 days. Fully differentiated cells were used for all experiments. All in vitro experiments were done in at least three biological replicates.

### Stromal vascular fraction isolation

For stromal vascular fraction (SVF) isolation, harvested sWAT depots from male C57Bl/6J mice were finely minced and digested in collagenase solution (2 mg/ml collagenase Type II, 10 mM CaCl2 and 0.5% free fatty acid free BSA in PBS) for 30 min at 37 °C at 16 × $g$. The digestion was stopped by adding 30 ml of full growth medium and digested solution was filtered through a 100 µm sieve. The SVF was pelleted by centrifugation at 600 × $g$ for 15 min. Afterwards, 30 ml of full growth medium was added and filtered through a 70 µm sieve (Corning, Arizona, United States). The SVF was pelleted by centrifugation at 600 × $g$ for 15 min und seeded in T75 flasks in growth medium (DMEM/F12 with glutamax supplemented with 10% FBS, 1% penicillin and streptomycin, (all Thermo Fisher Scientific, Waltham, MA, United States). For white adipocyte differentiation, fully confluent preadipocytes were treated with 1 µM dexamethasone, 0.5 mM 3-iso-butyl-1-methylxanthine, 1 µM rosiglitazone and 1.5 µg/ml insulin (all Sigma-Aldrich, St. Louis, MI, United States) for 4 days, and the differentiation medium was changed every other day. Afterwards, primary white adipocytes were kept in growth medium supplemented with 1.5 µg/ml insulin.

### CrispR/Cas9 clones

In total, 1 × 10⁶ C3H10T1/2 preadipocytes were seeded in a 6 well plate in antibiotic-free standard growth medium 72 h before transfection. At a confluency of 60% cells were transfected with 1.5 µg of the indicated plasmids (KO or control plasmid, Santa Cruz Biotechnology, Texas, United States) by using 10 µl UltraCruz transfection reagent (Santa Cruz Biotechnology, Texas, United States). The selection of positive clones was performed by growing the cells in standard growth medium supplemented with puromycin (Santa Cruz Biotechnology, Texas, United States; 2 µg/ml) for 10 days. RFP+ clones were separated via fluorescence activated cell sorting (FACS, by using FACS 7 Aria IIu (BD Biosciences, Franklin Lakes, United States)).

### Treatments

For starvation experiments, cells were carefully washed with PBS and maintained for 24 h in starvation medium (HBSS (Thermo Fisher Scientific, Waltham, MA, United States) supplemented with 10 mM HEPES (Thermo Fisher Scientific, Waltham, MA, United States)).

To stabilise p53 on protein level, iBACs were treated with 1 µM of the pharmacological compound Idasanutlin (Selleck Chemicals, Houston, United States) for 24 h.

For silencing experiments, fully differentiated SGBS cells were transfected with 100 nM of siCtrl or sip53 (Thermo Fisher Scientific, Waltham, MA, United States) by using Lipofectamine RNAiMax (Thermo Fisher Scientific, Waltham, MA, United States) following the provider's instructions. Cells were harvested 48 h after transfection either in growth medium or after 24 h in starvation medium.

For TP53-R72P overexpression experiments fully differentiated (day 7) C3H10T1/2 CrispR/Cas9 p53 KO adipocytes were detached using trypsin (0.25%)/collagenase (0.5 mg/ml), washed with PBS and resuspended in electroporation buffer R (Neon Transfection system, MKP1096, Invitrogen) to $6 \times 10^5$ cells per 10 μl. Control expression vector (V5-EGFP-miniTurbo-NES[27]), pcDNA3.Flag p53-WT P72 (kind gift of Tim Barnoud, Medical University of South Carolina) or pcDNA3.Flag p53-WT P72 were added to the cells at a concentration of $2 \mu g/6 \times 10^5$ cells. Electroporation was performed with a 10 μl Neon transfection tip at a voltage of 1300 V, pulse width of 20 ms, and 2 pulses. Three electroporation reactions were reseeded into one well of a 12-well plate containing antibiotic-free standard growth medium and allowed to recover for 24 h. The next day, cells were carefully washed with PBS and maintained in starvation medium (HBSS (Thermo Fisher Scientific, Waltham, MA, United States) supplemented with 10 mM HEPES (Thermo Fisher Scientific, Waltham, MA, United States)) for 24 h before isolating RNA for qPCR analysis.

## Mice

All animal studies were approved by the Austrian Ministry for Education, Science and Research (Vienna, Austria, BMWFW-66.010/0087-WF/V/3b/2017) and performed strictly according to its guidelines. Adiponectin-CreERT2 mice were a kind gift from Tim J. Schulz and licensed through MTA by Pierre Chambon. P53-Lox mice were originally obtained from the National Cancer Institute (NCI) Mouse Repository (Strain: FVB.129P2-Trp53tm1Brn/Nci; Stock No. 01XC2). To generate AdipoQ-CreERT2 x p53-Lox mice, female mice heterozygous for the CRE-transgene and homozygous for loxP-flanked allele were bred with male mice homozygous for loxP-flanked allele starting at the age of 10 weeks. All animals were housed in a controlled (22 °C, 40–60% relative humidity) environment with a 12:12 h light-dark cycle and allowed food and water ad libitum. As male mice more robustly develop symptoms of metabolic syndrome than female, only male Ctrl and KO mice were challenged with a 60% high fat diet (D12492, HFD, Research Diets, New Brunswick, USA) for 12 weeks, starting at the age of 5 weeks, before p53 ablation was facilitated by treating the mice with Tamoxifen (100 mg/kg via oral gavage for 5 consecutive days (Molekula group, Munich, Germany) followed by a 1 week wash out period. Afterwards, mice were either kept on an ad libitum HFD or submitted to a 4:3 intermittent fasting regime (4 days of ad libitum HFD, 3 days of water only fasting, 24 h each, 9:00–9:00) for 18 days. Metabolic phenotyping was performed starting after 1 week of intermittent fasting. Animals were sacrificed after the last 24 h of fasting. Standard housing involved group housing with bedding, if applicable with the experimental design.

## Human cohorts

Human plasma samples and biometric data of the fasting-mimicking diet (FMD) cohort were kindly provided by Alba Sulaj (Department of Endocrinology, Diabetology and Clinical Chemistry, Heidelberg University Hospital) with ethical approval from the University Hospital of Heidelberg (Ethic-Nr. S-682/2016). The randomised controlled open study (German Clinical Trials Register (Deutsches Register Klinischer Studien DRKS), DRKS-ID: DRKS00014287) was designed as previously published[84]. In short, patients with type II diabetes (6 females and 14 males, age 52–75) signed informed consents and were instructed to comply 5 consecutive days of FMD per month and to return to their normal diet until the next diet cycle, which was initiated about 25 days later. FMD is a plant based diet; day 1 of FMD supplied 4600 kJ (11%

protein, 43% carbohydrates and 46% fat), day 2–5 provided 3000 kJ (9% protein, 44% fat, and 47% carbohydrate) per day[85]. Blood samples and peripheral blood mononuclear cells of patients were collected in the fasted state and plasma samples were analysed before and after 3 cycles of the FMD intervention. For the bariatric surgery cohort, biometric data as well as visceral and subcutaneous AT samples from the Leipzig Obesity Biobank were kindly provided by Matthias Blüher (Department of Medicine, University of Leipzig). AT biopsies were obtained from 135 individuals (37 men, 98 women, age 16–70) with morbid obesity in the context of a two-step bariatric surgery approach[11], which in most cases included a sleeve gastrectomy as the first step and laparoscopic Roux-en-Y gastric bypass as second step. Adipose tissue samples were collected during elective laparoscopic abdominal surgery as described elsewhere[86], immediately frozen in liquid nitrogen and stored at −80 °C. Measurements of body composition and metabolic parameters were performed as previously described[87]. In short, BMI was calculated as weight divided by squared height. Weight loss re-gainers and weight loss-retainers were defined by a cut-off of 25% BMI loss. The study was performed in agreement with the Declaration of Helsinki and approved by the Ethics Committee of the University of Leipzig (approval number 159-12-21052012). All study participants gave written consent to use their data in an anonymized form for research purposes before taking part in this study.

## Organ harvesting

Mice were euthanised by cervical dislocation and tissues were flash-frozen in liquid nitrogen. Between 100 μl and 500 μl of blood was mixed with 10 μl of EDTA and kept at room temperature for about 30 min. Afterwards, samples were centrifuged at $1200 \times g$ at 4 °C for 10 min. Plasma was transferred to clean tubes and stored at −80 °C. Organ weight was analysed by using an analytical scale.

## Isolation of adipocyte-rich fraction

For adipocyte fractionation, freshly harvested tissue was finely minced with scissors and digested using Collagenase Type II (1 mg/ml, Thermo Fisher Scientific, Waltham, MA, United States) dissolved in PBS (Thermo Fisher Scientific, Waltham, MA, United States), supplemented with 10 mM CaCl2 (Sigma-Aldrich, St. Louis, MI, United States) and shaken (110 rpm) at 37 °C for 30–60 min. Enzymes were inactivated by the addition of DMEM containing 10% FBS. The upper layer containing mature adipocytes was filtered through a 100 μm cell strainer (Corning, Massachusetts, United States) and centrifuged at $100 \times g$ for 10 min. The mature adipocyte fraction was washed several times with PBS and lysed in 700 μl Qiazol (Sigma-Aldrich, St. Louis, MI, United States).

## Glucose and insulin tolerance tests

Mice were fasted for 6 h prior to glucose tolerance test and 4 h prior to insulin tolerance test, starting at 6:00 a.m. Mice were single-housed during the fasting period and the duration of the test. Water was provided ad libitum. Basal glucose levels were measured before intraperitoneal injection of the glucose or insulin solution (fixed dose of 42 mg glucose (Sigma-Aldrich, St. Louis, MI, United States), and 0.02 IU Insulin (Novo Nordisk, Bagsvaerd, Denmark)), respectively. Blood glucose levels (mg/dl, measured with Accucheck guide glucometer (Roche, Basel, Switzerland)) were measured after 15, 30, 60, and 120 min of glucose injection. Blood samples were taken from the tail vein.

## Metabolic cages

Before metabolic recording, mice were trained to use the drinking bottles of metabolic cages for 1 week and acclimated to metabolic cages for 48 h. Metabolic assessment was performed using an indirect calorimetry system (TSE PhenoMaster, TSE Systems, Bad Homburg, Germany). Animals were single-housed at room temperature, at a

regular light-dark (12:12 h) cycle, and with free access to food and water. For acute fasting experiments, food was withdrawn at 9:00 a.m. for 24 h, 1 week after starting the metabolic recording. For the diet-induced obesity cohort, food was withdrawn 3 days per week for 24 h each, with alternating ad libitum HFD feeding days. $O_2$ consumption, $CO_2$ production, and locomotor activity (using infrared sensor frames) were measured every 15 min.

## Blood chemistry

Plasma fatty acid, triglyceride, cholesterol, alanine transaminase and aspartate transaminase were analysed with the serum analyser (Beckman Coulter, AU480). Insulin levels were determined by using the Ultra-Sensitive Mouse Insulin ELISA Kit (CrystalChem, Zaandam, Netherlands) according to manufacturer's instructions. Ketone body measurements were analysed in tail vein blood (Wellion, Marz, Austria). Plasma TNFα levels were analysed by using the TNF alpha Mouse ProQuantum Immunoassay Kit (Thermo Fisher Scientific, Waltham, MA, United States) according to the manufacturer's instructions. Murine plasma proteome was analysed by using the Proteome Profiler Mouse Adipokine Array Kit (R&D systems, Minneapolis, Canada) according to the manuals.

## Tissue isolation

Flash-frozen tissues were lysed in Qiazol by using Magnalyser beads (PeqLab, Radnor, United States) at $4000 \times g$ for 20 s (2 runs) with the TissueLyser (Thermo Fisher Scientific, Waltham, MA, United States). In between runs, samples were cooled by incubating them on ice for 5 min. RNA was isolated by using the PeqGOLD total RNA kit (Peqlab, Radnor, United States) according to the manuals. RNA concentration and purification were quantified with NanoDrop® ND-1000 (Peqlab, Radnor, United States).

## Western blotting

For western blotting experiments, tissues were lysed with Magnalyser beads in radioimmuno-precipitation assay (RIPA) buffer (50 mM Tris-HCl, 150 mM NaCl, 2 mM EDTA, 50 mM NaF, 0.1% SDS, 0.5% Nadeoxycholate, 1% NP-40, adjusted to pH 7.2–7.4) supplemented with PIC (complete Tablets EASYpack, Roche, Basel, Switzerland) and PhosStop (Roche, Basel, Switzerland). Afterwards, samples were incubated on ice for 20 min, and cellular debris were pelleted at $15,000 \times g$ for 15 min. The protein concentration of cleared supernatants was analysed by using a bicinchoninic acid assay kit (BCA, Thermo Fisher Scientific, Waltham, MA, United States). Antibodies used: p53 (32532 (D2H9O)), 1:1000 Cell Signalling, Danvers, MA, United States), human p53 (sc-126 (DO-1), 1:3000, Santa Cruz Biotechnology, Dallas, Texas, United States), GAPDH (2118 (14C10), 1:5000, Cell Signalling, Danvers, MA, United States), β-actin (Ab6276 (AC-15), 1:250,000, Abcam, Cambridge, United Kingdom), vinculin (PA5-29688, 1:1000, Thermo Fisher Scientific, Waltham, MA, United States), MMP12 (ab52897 (EP1261Y), 1:1000, Abcam, Cambridge, United Kingdom). Goat anti-Rabbit IgG (31460, 1:5000, Thermo Fisher Scientific, Waltham, MA, United States) and Goat anti-Mouse IgG (31430, 1:5000, Thermo Fisher Scientific, Waltham, MA, United States) were used as secondary HRP-conjugated antibodies.

## Targeted sequencing of the TP53 SNP

DNA of white blood cells was isolated by using the DNeasy Blood and Tissue kit (Thermo Fisher Scientific, Waltham, MA, United States) according to manufacturer's instructions. For genotyping of the TP53 P72R SNP (rs1042522) 15–50 ng of DNA was amplified using M13 tailed primers spanning exon 4 (TP53_ex4_F: tgagtggatccattggaaggg, TP53_ex4_R: tccaaacaaaagaaatgcagggg) and the HotstarTaq PCR Mastermix (5 units/μl) (QIAGEN, Hilden, Germany). After purifying 5 μl of the PCR products (Thermo Fisher Scientific, Waltham, MA, USA) using enzymatic ExoSAP-ITTM cleanup, 1 μl was used for Sanger sequencing.

To this end, templates were prepared using 0.5 μl BigDye™ Terminator v3.1 Cycle Sequencing Kit and M13 universal primers (M13_F: CACGAC GTTGTAAAACGAC, M13_R: GGATAACAATTTCACACAGG). Sequencing reactions were purified using with SephadexTM G-50 Superfine (VWR, Radnor, PA, USA) and sequenced on an ABI 3730 instrument (Thermo Fisher Scientific). Data analysis was performed using SeqScape Software v3.0 (Thermo Fisher Scientific). Out of 20 patients, 14 were homozygous for the R72 variant, five were heterozygous for the P72 variant and one patient was homozygous for the P72 variant.

## qPCR analysis

Isolated RNA was reverse transcribed to cDNA by using High-Capacity cDNA Reverse Transcription Kit (Thermo Fisher Scientific, Waltham, MA, United States), before cDNA was amplified using Blue SybrGreen qPCR Mastermix (Biozym Scientific, Olendorf, Germany). Primer sequences are given in Supplementary Table 1.

## Cut and run protocol

The cut and run protocol was performed as previously described[88]. In short, nuclei of differentiated hMADs were isolated in lysis buffer containing digitonin. The nuclei were bound to activated Concanavalin beads and incubated with the primary antibody (1:100, DO-1, Santa Cruz Biotechnology, Dallas, United States) or isotype control overnight at 4 °C. After washing the beads, the secondary antibody was incubated for 1 h at 4 °C. To fragment the DNA, 1 ug/ml of pA-MNAse was added and incubated at 4 °C while rotating. Primer for target-gene regions were designed based on ChIP-validated p53 binding sides in the intronic regions. Negative control primers were designed at loci with no predicted p53 binding side. qPCR of eluted DNA fragments was performed according to a previously published protocol[89]. Briefly, eluted DNA fragments were purified (Qiagen MinElute purification kit) and amplified by qPCR. The fold enrichment was calculated by using the $2-\Delta\Delta$ CT method.

## Proteomics

Proteomics was performed as previously described[90]. In short, harvested eWAT depots were homogenised with RIPA buffer using metal beads and the LT TissueLyser (Thermo Fisher Scientific, Waltham, MA, United States). Lysed samples were sonicated, centrifuged at $19,000 \times g$ for 15 min and the protein concentration in the supernatant was quantified with BCA assay (Thermo Fisher Scientific, Waltham, MA, United States). Proteins were precipitated with methanol/chloroform. The pellet containing the proteins was reconstituted with 100 mM Tris (pH 8.5) and 2% sodium deoxycholate and alkylated by adding 5 mM TCEP and 30 mM chloroacetamide at 56 °C for 10 min. Afterwards, proteins were lysed with 1:100 Lys-C and 1:50 trypsin at 37 °C overnight and lysis was stopped by adding 1% trifluoroacetic acid to a final concentration of 0.5%. Digested peptides were desalted on an Oasis HLB plate (Waters, Massachusetts, United States) and dried, before adding 2% formic acid. Peptides were subjected to liquid chromatography-tandem mass spectrometry (MS/MS) analysis. A total of 1000 ng of tryptic peptides was analysed using an Ultimate3000 high-performance liquid chromatography system (Thermo Fisher Scientific, Waltham, MA, United States) coupled to a Q Exactive HF-x mass spectrometer (Thermo Fisher Scientific, Waltham, MA, United States). Buffer A was water acidified with 0.1% formic acid; buffer B consisted of 80% acetonitrile and 20% water with 0.1% formic acid. Peptides were trapped for 1 min at 30 μl/min with 100% buffer A (0.3 mm by 5 mm with PepMap C18, 5 μm, 100 Å; Thermo Fisher Scientific, Waltham, MA, United States). Afterwards, peptides were separated by a 50-cm analytical column packed with C18 beads (Poroshell 120 EC-C18, 2.7 μm; Agilent Technologies, California, United States). The gradient was 9 to 40% B in 155 min at 400 nl/min. Buffer B was then raised to 55% in 10 min and elevated to 99% for the cleaning step. Peptides were ionised using a spray voltage of 1.9 kV and a capillary heated at 275 °C. The

mass spectrometer was set to acquire full-scan MS spectra (350 to 1400 mass/charge ratio) for a maximum injection time of 120 ms at a mass resolution of 120,000 and an automated gain control (AGC) target value of $3 \times 10^6$. Up to 25 of the most intense precursor ions were selected for MS/MS. HCD fragmentation was performed in the HCD cell, with the readout in the Orbitrap mass analyser at a resolution of 15,000 (isolation window of 1.4 Th) and an AGC target value of $1 \times 10^5$ with a maximum injection time of 25 ms and a normalised collision energy of 27%.

### Proteomic data analysis

Raw data were analysed by using MaxQuant v1.6.17 software using the integrated Andromeda search engine and the Mouse UniProt Reference Proteome. MaxQuant analysis was performed by using the standard parameters (the "label-free quantification" and "match between runs" were selected with automatic values) with only the addition of deamidation (N) as variable modification. Data analysis was done with Perseus v1.6.14. The proteins were filtered for reverse, potential contaminants and identified by site. Label-free quantification was calculated by MaxQuant and missing values were imputed by Perseus using the automatic settings (width, 0.3; down shift, 1.8; mode, separately for each column). The $z$-score (mean per row) was calculated, ANOVA testing (Benjamini–Hochberg FDR, 0.05) was performed to analyse statistical significance, and hierarchical clustering was performed. Pathway analysis of differentially expressed proteins in each cluster was performed by using the enrichR package (Wilcoxon Rank Sum test with Bonferroni multiple testing correction).

### Lipolysis assay

Lipolysis assay of ex vivo explants has been performed as previously described[91]. In short, small eWAT or sWAT pieces were incubated in 200 μl of DMEM high glucose supplemented with 2% FA-free BSA for 30 min at 37 °C. Afterwards explants were transferred to fresh medium and incubated at 37 °C for 1 h. To stimulate lipolysis, explants were transferred to full medium supplemented with 10 μM of isoproterenol (Sigma-Aldrich, St. Louis, MI, United States) for 1 h. NEFA release was analysed by using a colorimetric assay (Fujifilm Wako, Neuss, Germany) and normalised to the protein concentration of explants.

### Nuclei isolation

eWAT depots were immediately flash frozen in liquid nitrogen after harvesting. Frozen tissues were thawn, carefully minced in lysis Buffer (10 mM Tris-Hcl, 146 mM NaCl, 21 mM MgCl$_2$, 1 mM CaCl$_2$, 0.1% CHAPs pH = 7.4) and grinded by using a tight douncer (Thermo Fisher Scientific, Waltham, MA, United States) on ice. Homogenised tissue was filtered through a 40 μm sieve (Corning, Massachusetts, United States), centrifuged at $500 \times g$ for 5 min at 4 °C to pellet the nuclei and resuspended in buffer supplemented with 1% BSA and 0.2 U/μl RNAase inhibitor (Takara Bio, Kusatsu, Japan).

### 10X library preparation and sequencing

Nuclei were isolated from a pool of eWAT depots from 3–4 mice (HFD-AL: $n = 3$; HFD-IF: $n = 4$, HFD-IF-KO: $n = 3$) and aggregated into one snRNAseq library per group. 10X-based libraries were generated following manufacture's protocol (10X Genomics, Pleasanton, CA, United States). Briefly, 1000 nuclei/μl suspension was loaded to 10X chromium with a V3 kit. Sequencing was performed using a S1 Novaseq Flowcell (Illumina, Eindhoven, Netherlands). The number of nuclei identified after cleaning the dataset in the single libraries are 12841, 14260, and 18238 nuclei in the HFD-AL, HFD-IF, and HFD-IF-KO group respectively and the median gene count per cell was 2087 (HFD-AL), 2198 (HFD-IF), and 1738 (HFD-IF-KO).

### Single cell data analysis

For data analysis, CellBender[92] was used to distinguish cell-containing droplets from empty droplets and DoubletFinder[93] was used to exclude potential droplets. The Seurat pipeline was used for batch-correction, clustering, signature gene identification and differential gene expression analysis. Ligand-receptor interactions were analysed by using Cellchat[94]. Overrepresentation analysis of differentially de-repressed genes in the adipocyte cluster was performed using WebGestalt mapping to gene ontology–biological processes (GO-BPnr). Differentially expressed genes in Supplementary Data 1 and 2 were derived with the FindMarkers function in Seurat using a Wilcoxon Rank Sum test with Bonferroni multiple testing correction and avgLog2FC (>0.5) and adjusted $p$ value (<0.01) cut-offs.

### RNA-seq analysis of human adipose tissue samples and cells

rRNA-depleted RNA-seq data were prepared on the basis of the SMARTseq protocol[95,96]. All libraries were sequenced on a Novaseq 6000 instrument at Functional Genomics Center Zurich. Adaptor and low-quality reads were trimmed using Fastp (v.0.20.0)[97] taking into account a minimum read length of 18 nts and a quality cut-off of 20. Read alignment to the human reference genome (build GRCh38.p13) and gene level expression quantification (gene model definition from GENCODE release 32) was carried out using Kallisto (v0.46)[98]. Human adipose tissue cohort was split into responders and non-responders based on at least 25% BMI loss between pre- and post-surgery. Two-tailed paired $t$-test with multiple testing adjustment (Bonferroni correction) was applied to the normalised gene counts (transcripts per millions) of responders and non-responders between the two time points. For human SGBS cells RNA was prepared as described above. After quality control (RIN > 7) samples were oligo-dT enriched, fragmented, reverse transcribed, submitted to strand-specific mRNA library preparation, and sequenced with DNA nanoball technology (DNBSEQ, BGI). Data were aligned to the human hg38 built (UCSC) and analysed with the Dr.Tom data visualisation solution (BGI) that includes gene set enrichment analysis (GSEA) for gene ontology and KEGG pathways (FDR < 0.05).

### Histology

Immunohistochemical stainings of formalin-fixed, paraffin embedded tissues were performed after antigen retrieval (93 °C, 15 min at pH 6) and peroxidase blocking (Agilent, Foster City, CA, United States) using the UltraVision LP detection system (Thermo Fisher Scientific, Waltham, MA, United States) according to the manual for cleaved caspase 3 antibody (9661, 1:50, Cell Signalling Technology, Denver, MA, USA) or Ki67 antibody (12202, 1:400, Cell Signalling Technology, Denver, MA, USA).

AEC (3-amino-9-ethyl carbazole) chromogen (Thermo Fisher Scientific, Waltham, MA, United States) was used for colour detection. Counterstaining with hematoxylin was done on all slides.

White adipocyte cell size was quantified by using an automated plugin for ImageJ (Adiposoft[99]) and quantification of crown-like structures and cleaved caspase 3 positive cells was performed by using ImageJ. Liver lipid droplets in H&E sections were counted using VisioPharm version 2019.09.

### Electron microscopy

Electron microscopy of eWAT samples ($n = 3$ for each HFD-IF and HFD-IF-KO groups) was performed according to a previously published protocol[100]. In short, eWAT was dissected and the bulk tissue immediately flash frozen in liquid nitrogen without cryo-preserving agents. Frozen tissue samples were cut into ~5 mm$^3$ pieces and fixated at 37 °C in 2.5% (wt/vol) glutaraldehyde and 2% (wt/vol) paraformaldehyde. Imaging was performed after dehydrating, embedding, and trimming of the samples. Up to 10 electron micrographs were taken per eWAT

from each mouse using a scanning electron microscope (Zeiss Sigma 500). Scanning transmission electron microscopy (STEM) imaging mode of a field emission scanning electron microscope (ZEISS FE-SEM Sigma 500) in combination with ATLAS TM was used to perform imaging on large areas of eWAT material with high resolution.

## Automated image analysis and quantification

Image analysis was performed using the image analysis software Visiopharm version 2019.09. Tissue area in whole tissue scans (Olympus V200) was detected using a customised, pretrained tissue detection app for brightfield images and corrected manually. Adipocytes were detected based on a polynomial local linear feature for identifying cell contours and a colour intensity threshold. Adipocytes size distribution was classified into 18 size classes ranging from diameters of 20 to 200 μm based on the object area, which was used to calculate the diameter of a circle with the same area. Non-round adipocytes were excluded. For Ki67 immunohistochemistry quantification, nuclei were detected using the app *Nuclei Detection, AI (Brightfield)* by Visiopharm. The nuclei were then classified as Ki67 positive based on a threshold above 90 (intensity range 0–255) over each whole nucleus. Intensities were based on a staining feature for the Ki67 stain defined within Visiopharm. Crown-like structures were detected based on the surrounding image areas of identified adipocytes, with the following criteria: detection of nuclei, haematoxylin intensities above an intensity threshold of 80 (intensity range 0-255), and a diameter of ≥10 μm.

## Quantification and statistical analysis

All data are presented as the mean ± SEM unless otherwise indicated in the figure legend. Statistically significant differences were tested as described in the figure legends using GraphPad Prism version 9. $*p < 0.05$, $**p < 0.01$, and $***p < 0.001$ with 95% confidence interval. Differences not indicated with asterisks or indicated with ns are not statistically significant ($p > 0.05$). For the human bariatric surgery cohort, outliers were detected with a Grubbs' test and removed.

## Reporting summary

Further information on research design is available in the Nature Portfolio Reporting Summary linked to this article.

## Data availability

The sequencing data generated in this study have been deposited in the Gene Expression Omnibus (GEO) repository under accession code GSE229690 for the snRNA-Seq dataset and GSE248893 for the RNA-sequencing dataset of SGBS adipocytes. The proteomic data were deposited at PRIDE repository under the accession number PXD041351. Any additional information required to reanalyse the data reported in this paper is available from the lead contact upon request. Source data are provided with this paper.

## Materials availability

Information or requests for resources and reagents should be directed to and will be fulfilled by the lead contact, A.P. (E-mail: andreas.prokesch@medunigraz.at). AdipoQ-CreERT2 x p53-Lox mice will be available via the lead contact. Other than these, this study did not produce new and unique reagents or mouse models.

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

## Acknowledgements

We thank Prof. Pierre Chambon for making AdipoQ-CreERT2 mice available to us. We also thank Ass.Prof. Tim Barnoud (Medical University of Southern Carolina) for sharing P72 and R72 p53 overexpression vectors. At the Division of Cell Biology, Histology, and Embryology we are grateful to Prof. Berthold Huppertz for providing personnel support and Daniel Kummer for support with histological quantifications using VisioPharm. I.R. was funded by the PhD program Molecular Medicine at the Medical University of Graz and by a Marietta Blau-stipend. A.P. and D.K. were supported by the Austrian Science Fund FWF (P34109, P29328, I3165 to A.P., and F73 and P32400 to D.K.). D.K. was supported by the Province of Styria and the City of Graz. M.S. and T.J.S. were supported by a grant from the German Research Foundation (DFG; project number 323196138). A.S. was supported by the German Research Foundation (DFG project number 236360313–SFB 1118) and by the German Center for Diabetes Research (DZD project number 82DZD07C2G). The fasting-mimicking diet used in this study was funded by L-Nutra. L-Nutra has no role in the design or conduct of the study nor in the preparation, review, or approval of the manuscript. S.H. received funding by the SFB1118 (DFG) and the Helmholtz Future Topic AmPro. Parts of this work were funded by grants from the DFG (German Research Foundation)—Projektnummer 209933838—SFB 1052 (project B1 to M.B.) and by Deutsches Zentrum für Diabetesforschung (DZD, Grant: 82DZD00601 to M.B.). This work has also been supported by EPIC-XS (project number 368) funded by the Horizon 2020 program of the European Union. For open access purposes, the authors have applied a CC BY public copyright license to any author accepted manuscript version arising from this submission. Figure 8 was created with BioRender (www.biorender.com).

## Author contributions

I.R. designed and performed the experiments, analysed the data, and wrote the manuscript. H.M., E.M., J.K., R.X., Z.R., M.G., N.V., M.A., R.Z.C., L.C.H., and L.G. performed and assisted with experiments, and analysed data. T.W., M.W., D.Ko., A.Georgiadi, E.H., T.J.S., M.S., W.S., H.D., A.Ghosh, A.H., D.Kr., A.J.R.H., A.S., f.v.M., M.B., S.H., and C.W. contributed materials and data, and provided expertise and feedback. A.P. designed, coordinated, and supervised the project, analysed data, and wrote the manuscript.

## Competing interests

The authors declare no competing interests.

## Additional information

[1]Gottfried Schatz Research Center for Cell Signaling, Metabolism and Aging, Division of Cell Biology, Histology and Embryology, Medical University of Graz, Graz, Austria. [2]Institute of Food Nutrition and Health, Department of Health Sciences and Technology, Eidgenössische Technische Hochschule Zürich (ETH), Schwerzenbach, Switzerland. [3]Institute for Diabetes and Cancer, Helmholtz Munich, German Center for Diabetes Research (DZD), Neuherberg, Germany. [4]Department of Endocrinology, Diabetology, Metabolism and Clinical Chemistry (Internal Medicine 1), Heidelberg University Hospital, Heidelberg, Germany. [5]Gottfried Schatz Research Center, Molecular Biology and Biochemistry, Medical University of Graz, Graz, Austria. [6]Biomolecular Mass Spectrometry and Proteomics, Bijvoet Center for Biomolecular Research and Utrecht Institute of Pharmaceutical Sciences, Utrecht University, Utrecht, The Netherlands. [7]Division of Pediatric Endocrinology and Diabetes, Department of Pediatrics and Adolescent Medicine, University Medical Center Ulm, Ulm, Germany. [8]Core Facility Ultrastructure Analysis, Medical University of Graz, Graz, Austria. [9]Institute of Human Genetics, Diagnostic and Research Center for Molecular BioMedicine, Medical University of Graz, Graz, Austria. [10]Department of Adipocyte Development and Nutrition, German Institute of Human Nutrition, Nuthetal, Germany. [11]German Center for Diabetes Research (DZD), München-Neuherberg, Germany. [12]University of Potsdam, Institute of Nutritional Science, Nuthetal, Germany. [13]Institute of Pharmacology, Max Rubner Center (MRC) for Cardiovascular Metabolic Renal Research, Charité-Universitätsmedizin Berlin, Corporate Member of Freie Universität Berlin, Humboldt-Universität zu Berlin, Berlin, Germany. [14]Department of Bioengineering, Stanford University, Stanford, CA, USA. [15]Stem Cell Biology and Regenerative Medicine Institute, University of Stanford, Stanford, CA, USA. [16]Functional Genomics Center Zurich, Eidgenössische Technische Hochschule Zürich (ETH), Zurich, Switzerland. [17]Helmholtz Institute for Metabolic Obesity and Vascular Research (HI-MAG) of the Helmholtz Center Munich at the University of Leipzig and University Hospital Leipzig, Leipzig, Germany. [18]BioTechMed-Graz, Graz, Austria. [19]Laboratory of Nutrition and Metabolic Epigenetics, Institute for Food, Nutrition and Health, Department of Health Sciences and Technology, ETH Zurich, Zurich, Switzerland. [20]Department of Medicine, University of Leipzig, Leipzig, Germany. ✉e-mail: andreas.prokesch@medunigraz.at

