## [Peer Review File · Nature Communications]

Adipocyte p53 coordinates the response to intermittent fasting by regulating adipose tissue immune cell landscapeREVIEWER COMMENTS

Reviewer #1 (Remarks to the Author):

This study investigates the role of adipocyte p53 in lipid-associated macrophage in adipose tissue under intermittent fasting. The results in preclinical models and human are novel and of interest since our knowledge on the mechanisms underlying the action of intermittent fasting remain largely elusive. The hypothesis is original, the manuscript is easy to follow and the conclusions are supported by the findings.

Comments

-It is known that p53 can be regulated at post-transcriptional level. I would recommend to measure protein levels of p53 at least in HFD-AL vs HFD-IF.

-The current results show extensive transcriptomic results, but the main question is whether those changes in RNA are consequence of a direct or indirect interaction with p53. ChiPseq would be very interesting or at least, to prove the direct interaction between p53 and some of the identified genes (Angptl4 could be really relevant) would be valuable for the study.

-Have the authors studied whether the lack of p53 in adipocytes could affect cell cycle and or proliferation/apoptosis or have they observed any tumor development?

-Human data are exciting and of great interest. I would recommend to investigate the effect of p53 variants (P72 y R72) at least in in vitro experiments.

Minor:

-Fig 1F. Why the n is lower in this fig than in the other graphs?

Reviewer #2 (Remarks to the Author):

1. How many mice were pooled together as one snRNAseq sample? How many samples were sequenced for each group? How many cells were identified in each sample? How many genes were identified?
2. The mechanisms through which IF induces P53 expression in adipocytes should at least be discussed.
3. The authors claimed that IF induces adipocyte death but yet also causes an improvement in insulin sensitivity. Why?
4. The analysis of scRNAseq data is relatively superficial. Differential gene expression should be performed for each major cell type (such as adipocyte, FAP) and major immune cell types (such as LAM) among different treatment groups. The expression of genes/protein listed in Figures 1, 2, and 4 should be checked in snRNAseq. Is the expression of Tp53 and relevant genes altered in other cell types?
5. Please double check the label of the color scale in Figure 3F.

Reviewer #3 (Remarks to the Author):

The current manuscript identifies that P53 in adipocytes plays a key role in induced metabolic remodeling in response to intermittent fasting (IF). The authors observed that IF induces macrophage infiltration in adipose tissues, coinciding with an increase in the expression of apoptosis genes, including P53 in adipocytes. snRNA-seq analysis reveals that IF has a major impact on adipose cell composition. While the number of adipocytes is greatly diminished, the number of immune cells, particularly lipid-associated macrophages (LAMs), is increased. This effect was largely reversed by adipocyte-specific deletion of p53. Adipocyte-specific p53 KO mice showed improved metabolic flexibility in response to IF. The overall studies are well-conducted, and the mouse genetics data are very convincing. The use of snRNA-seq of whole fat is also appropriate. I address the following points to strengthen the manuscript.

1. One of the key observations of this study is that P53 KO adipocytes show increased catabolic responses rather than undergoing apoptosis. It will be important whether P53 has cell-autonomous regulation in lipolysis and β -oxidation. Do the DEGs or KEGG pathway involved in lipid metabolism emerge in p53 KO adipocytes?

2. Human data from P53 variants are very intriguing. However, it needs to be addressed whether the two variants have a correlation with P53 expression levels.

3. Additional analysis needs to be done to better understand adipose histology.

- Adipocyte number and size analysis need to be conducted in control, HFD-AL, HFD-IF, and HFD-IF-KO WAT.

- Is the expression of P53 in adipocytes a reversible process? How is the P53 level affected in the stromal vascular fraction affected by IF?

4. Cell type interaction analysis from snRNA-seq data also showed a profound effect on the interaction between immune cells and endothelial cells. Do they have potential role in P53-dependent remodeling?

Response to reviewers' comments:

Reviewer #1 (Remarks to the Author):

This study investigates the role of adipocyte p53 in lipid-associated macrophage in adipose tissue under intermittent fasting. The results in preclinical models and human are novel and of interest since our knowledge on the mechanisms underlying the action of intermittent fasting remain largely elusive. The hypothesis is original, the manuscript is easy to follow and the conclusions are supported by the findings.

Comments

-It is known that p53 can be regulated at post-transcriptional level. I would recommend to measure protein levels of p53 at least in HFD-AL vs HFD-IF.

We thank the reviewer for this comment.

Accordingly, in new *in vitro* experiments in C3H10T1/2 adipocytes, we show that p53 protein is induced by starvation conditions (**Fig rev 1.1** and **new Fig 2j**). Furthermore, the starvation-dependent increase in p53 protein returned to basal levels within 24 hours of refeeding (**Fig rev 1.2= new Supplementary Fig. 2g**). This data was also corroborated on the mRNA level in C3H10T1/2 (**Fig rev 1.3 = new Supplementary Fig. 2h**) and human hMADs adipocytes (**Fig rev 1.4**), indicating that p53 signalling is nutrient sensitive in human and mouse adipocytes.

Furthermore, we performed western blot in the cohorts from the original manuscript. Consistent with not detecting p53 in our bulk eWAT proteomics data, it was not possible to detect p53 by western blot in bulk eWAT in all cohorts (example shown for our main mouse cohort in **Fig rev 1.5**). Hence, we assume that in samples from mouse adipose tissue, methods assessing protein are not sensitive enough and/or are complicated by the fact that adipocytes comprise only a small fraction of cells in bulk eWAT. Furthermore, isolating the adipocyte-rich fraction did not yield enough protein to perform western blotting.

Together, we can detect dynamics in p53 protein in cultured adipocytes and on the level of target gene expression (see also throughout first version of manuscript).

Fig rev 1.1 = new Fig. 2j. Differentiated mature C3H10T1/2 adipocytes with wild-type (WT) or knocked out KO p53 were starved (HBSS/HEPES) for 24 hours or kept in full

medium for the same time (Ctrl). p53 antibody (D2H9O, Cell Signaling) was used to probe blots. GAPDH serves as loading control.

Fig rev 1.2 = new Supplementary Fig. 2g. Differentiated mature C3H10T1/2 adipocytes which were starved (HBSS/HEPES) for 24 hours, refeed for the indicated times or kept in full medium for the same time (Ctrl). p53 antibody (D2H9O, Cell Signaling) was used to probe blots. GAPDH serves as loading control.

Fig rev 1.3 = new Supplementary Fig. 2h. Differentiated mature C3H10T1/2 adipocytes were starved (STV; HBSS/HEPES) for 24 hours and refeed with normal medium as in control for the indicated times. Canonical p53 target gene expression was measured by qPCR (n=4, * and \$ denote significant difference to control and to STV, respectively).

Fig rev 1.4. Differentiated mature hMADs adipocytes were starved (HBSS/HEPES) for 24 hours and refed for the indicated times. Canonical p53 target gene expression was measured by qPCR.

Fig rev 1.5. p53 antibody (D2H9O, Cell Signaling) was used to probe blots with bulk eWAT samples from our main cohort. C3H10T1/2 cells with wild-type (WT) or knocked out (KO) p53 were used as positive control.

-The current results show extensive transcriptomic results, but the main question is whether those changes in RNA are consequence of a direct or indirect interaction with p53. ChiPseq would be very interesting or at least, to prove the direct interaction between p53 and some of the identified genes (Angptl4 could be really relevant) would be valuable for the study.

To test direct binding of p53 in target gene regions we conducted cut&run experiments in differentiated human adipocytes (hMADs). We confirmed an enrichment of predicted p53 binding sides in intronic regions of Angptl4 over isotype control, whereas negative control loci (Fam49a) were unchanged. These results are shown in **Fig rev 1.6** and in **new Supplementary Fig 4f** of the revised manuscript and suggest as tentative previously undescribed direct p53 targets in adipocytes.

Fig rev 1.6 = new Supplementary Fig 4f. Fold enrichment (target region over IgG isotype control) determined by CUT&RUN followed by qPCR targeting predicted p53 binding sides in an intronic region of Angptl4. Negative control primers (Fam49a) are designed at genomic regions that are remote from the targeted genomic region (n = 3).

Additionally, we investigated the dynamics of Angptl4 expression in response to nutritional challenges in differentiated C3H10T1/2 cells exposed to starvation and refeeding. The mRNA abundance of Angptl4 was strongly induced after 24 hours of starvation, and quickly returned to basal levels within the first 3 hours of refeeding (Fig rev 1.7= new Supplementary Fig. 4d). Thus, this data suggests that the expression of Angptl4 is dynamically regulated in response to nutrient deprivation and strongly correlates with the expression pattern of previously described p53 target genes (see Fig rev 1.3).

Fig rev 1.7 = new Supplementary Fig. 4e. Differentiated mature C3H10T1/2 adipocytes were starved (STV; HBSS/HEPES) for 24 hours and refeed with normal medium as in control for the indicated times. Angptl4 gene expression was measured by qPCR (n=4, * and \$ denote significant difference to control and to STV, respectively).

Furthermore, to estimate the transcriptional impact of activation of p53 signalling in adipocytes, we performed RNA-seq from starved human differentiated adipocytes (SGBS) with and without siRNA-mediated p53 knock down. Using GSEA hallmark analysis comparing the siTP53 to siCtrl group, we found de-enrichment of p53 signalling (containing for example p53 and Mdm2), apoptotic (containing Bax, Caspases and BCL2 family members) and inflammatory response (including Ccl2, Icam1, and inflammasome components) pathways, while cell cycle-related terms are found as enriched in the siTrp53 group (i.e. de-repressed by p53 knock down). Activation of apoptosis and induction of cell cycle arrest are well described functions of the p53 signalling pathway, although mostly for cancer cells (e.g. PMID: 33518400). However, p53 regulation of immune responses in adipocytes is less well known and in line with our in vivo data where adipocyte-specific p53 KO leads to reduced LAM abundance under IF. The RNA-seq results are shown in Fig rev 1.8 and as new Supplementary Fig. 2i, 4a, and 4b of the revised manuscript.

Fig rev 1.8 = new Supplementary Fig. 2i. GSEA hallmark analysis of RNA-seq data from starved, differentiated SGBS adipocytes with siRNA-mediated knock down of p53 (siTP53) compared to scrambled siRNA control (siCtrl). NES, normalized enrichment score; FDR, false positive rate.

-Have the authors studied whether the lack of p53 in adipocytes could affect cell cycle and or proliferation/apoptosis or have they observed any tumor development?

As mentioned above and shown in Fig rev 1.8 in our new RNA-seq data of starved, differentiated SGBS cells, cell cycle-related terms are significantly enriched in the siTP53 group, indicating that p53 represses cell cycle progression in starved adipocytes as is well known for cancer cells (e.g. PMID: 33518400). To also test this in vivo, we performed Ki67 staining of eWAT from our initial cohorts (**Fig rev 1.9 and new Supplementary Fig. 3h, i**). Quantifying these stainings shows a strong trend to higher Ki67-positive nuclei within crown-like structures comparing chow versus HFD-IF conditions. However, in adipocytes with knocked out p53, Ki67 staining trends to decrease in CLS and is unchanged in non-CLS regions when compared to HFD-IF conditions. As it is difficult to distinguish between adipocyte and non-adipocyte nuclei in histological sections, the results could reflect both proliferating AT-resident immune cells and mitotic adipocytes. To better differentiate, we measured cell cycle and proliferation markers in adipocyte-rich fractions from our cohorts (**Fig rev 1.10 new Supplementary Fig. 3j**). Marker gene expression showed a general trend to increase upon IF. This trend was reversed, at least in part, in the HFD-IF-KO group, confirming our Ki67 quantifications, but opposing the data from cultured adipocytes. This could be explained by the lack of interaction of adipocytes with immune cells in culture, that could emphasize p53 intrinsic effects rather than the cell-cell interaction within the tissue microenvironment.

Fig rev 1.9 = new Supplementary Fig. 3h, i. Ki67 positive nuclei within and outside of crown-like structures (CLS) in whole tissue eWAT sections as quantified with VisioPharm in CHOW, HFD-IF, and HFD-IF-KO groups. **(left panel)** Representative images from the 3 groups. Right panels: Ki67 staining with haematoxylin counter staining. Left images: Quantification results showing Ki67-negative (blue) and -positive (red) nuclei; Pink hatched areas are identified CLS. (scale bar = 50 μ m) **(right panel)** Fraction of Ki67 positive nuclei in whole section quantified in and outside of CLS.

Fig rev 1.10 = new Supplementary Fig. 3j. mRNA expression of markers encoding for proliferation-associated genes in the adipocyte-rich fraction of eWAT of HFD-AL, HFD-IF and HFD-IF-KO mice.

Finally, in our histological whole slide scans we could not detect any signs of tumour development in adipose tissue. This is consistent with the fact that liposarcomas are very rare (PMID: 37843627) and that the whole body p53 KO mouse is tumour free for several months, mainly developing lymphomas after 6 and more months (PMID: 1552940). Hence, our system with adipocyte-specific and inducible p53 knock-out (less than 5 weeks of KO, see Fig 3A) very likely precludes any tumour formation.

-Human data are exciting and of great interest. I would recommend to investigate the effect of p53 variants (P72 y R72) at least in in vitro experiments.

According to this reviewer's suggestion we cloned human p53 harbouring either the P72 or R72 variant and overexpressed them in C3H10T1/2 cells with CRISPR-mediated p53 KO. These cells were then starved to compare effects of the two variants in nutrient depletion conditions. As initially reported by the group of Maureen Murphy (PMID: 26947067, PMID: 28475405), we detect a differential effect of the P72 and R72 variant on the expression of Ccl2 (**Fig rev 1.11 and = new Supplementary Fig. 7d, e**), a gene encoding for a cytokine that is involved in monocyte chemotaxis. This is in line with earlier reports that show that Ccl2 is a p53 target (PMID: 26947067). Furthermore, the variants activate different downstream pathways in response to AMPK induction (PMID: 28475405). AMPK is a major fasting-response pathway, known to activate p53 from work from our lab (PMID: 27811061) and that of others (PMID: 15866171). In contrast to a strong induction of apoptosis of the P72 variant in response to starvation-mimicking conditions, the R72 variant preferentially induced senescence (PMID: 28475405). This data suggest that the R72 variant induces a more homeostatic transcriptional program in response to metabolic stress, which is consistent with our data in human FMD cohort that showed improved fasting response in patients carrying the R72 variant (Fig 7). This new data is shown in revised **Fig S7** and described in the revised discussion.

Fig rev 1.11= new Supplementary Fig. 7d, e. Differentiated mouse C3H10T1/2 adipocytes with CRISPR-mediated p53 KO were used to overexpress human TP53 harbouring either the R72 or the P72 variant using electroporation. 24 hours after electroporation cells were kept in starvation medium (STV; HBSS+HEPES) for another 24 hours. Left panel shows equal overexpression from the 2 plasmids. Right panel shows significantly reduced Ccl2 mRNA expression in samples overexpressing the R72 variant

(averaged expression from three independent electroporation experiments, one-sample t-test).

Minor:

-Fig 1F. Why the n is lower in this fig than in the other graphs?

The data in Fig 1f is derived from metabolic cages and we only submitted a subset of our cohorts to these analyses.

Reviewer #2 (Remarks to the Author):

1. How many mice were pooled together as one snRNAseq sample? How many samples were sequenced for each group? How many cells were identified in each sample? How many genes were identified?

Nuclei were isolated from a pool of eWAT depots from 3-4 mice (HFD-AL: n= 3; HFD-IF: n= 4, HFD-IF-KO: n= 3) and aggregated into one snRNAseq library per group. We apologize for not providing this information and now amended it in the methods section.

The number of nuclei identified after cleaning the dataset in the single libraries are given in Fig 3B and 4B (12841, 14260, and 18238 nuclei in the HFD-AL, HFD-IF, and HFD-IF-KO group, respectively) and the median gene count per cell was 2087 (HFD-AL), 2198 (HFD-IF), and 1738 (HFD-IF-KO). This information has been added to the revised manuscript.

The genes identified as differentially expressed (\log_2 FC = +/-0.5, adj p-value<0.01) in the main clusters are now given as **Supplementary Table 1 and 2**. Pathway analysis indicates that in adipocytes catabolism-related pathways are de-repressed by p53 knock out (**Fig rev 2.1 = new Fig. 6a**).

Fig rev 2.1 = new Fig. 6a. Differentially upregulated genes (IF-KO vs IF) in the adipocyte annotated cell cluster from snRNAseq data was mapped to gene ontology biological process non-redundant (GOBPnr) using WebGestalt. Enrichment ratio is given on the x-axis and FDR values are given in bars.

2. The mechanisms through which IF induces P53 expression in adipocytes should at least be discussed.

It has been previously shown that p53 signalling is induced in genetical- and diet-induced obesity models and in humans (e.g. PMID: 19718037). Our prior work established p53 as a regulator of the fasting response in adipose tissue (PMID: 24191950) and liver (PMID: 24191950, PMID: 27811061, PMID: 35868560). In liver models, we could show that p53 induction under starvation is dependent on AMPK signalling (PMID: 27811061), as was shown before in cancer cells (PMID: 15866171). From our current work we could surmise that fasting-activated AMPK signalling in adipocytes (PMID: 37673765) further aggravates the already heightened p53 signalling under HFD (PMID: 19718037). Additionally, the cycling between fasting and refeeding could constitute cellular stresses in the adipose

tissue, leading to p53 induction as a known stress sensor (PMID: 26122615). These stresses could stem from repeated shrinkage and enlargement of adipocytes causing mechanical stress against the less flexible extra-cellular matrix (PMID: 37855264, PMID: 35625493). The inflammatory phenotype under IF could, therefore be a consequence and/or a contributor to this stress scenario.

We thank the reviewer for raising this point and now include these possible explanations in the discussion of the revised manuscript.

3. The authors claimed that IF induces adipocyte death but yet also causes an improvement in insulin sensitivity. Why?

We thank the reviewer for this important question. There are several adipose-independent effects of cyclic fasting and refeeding on whole-body energy homeostasis that ameliorate insulin sensitivity. In addition to overall weight loss and concomitant reduction in plasma glucose levels, these mechanisms include the increased secretion of incretins during refeeding (PMID: 31392745) and reduction of circulating growth factors such as liver-secreted IGF during the fasting phase (PMID: 35487190). To address this reviewer's question in our model we investigated the liver phenotype in our cohorts during the revision of our manuscript. As reported by others (e.g. PMID: 28918936), IF reduced liver fat deposition (**Fig rev 2.2 and = new Supplementary Fig. 2f**) and led to a trending reduction of inflammatory and macrophage markers on the HFD background (**Fig rev 2.3**). This reinforces that other major metabolic organs respond differently to the IF regimen and that improvement of fatty liver and liver inflammation status may contribute to promote systemic health. These new data are provided in Fig S1 and treated in the discussion of the revised manuscript.

Fig rev 2.2 = new Supplementary Fig. 2f. Histologic analysis of livers of HFD-AL and HFD-IF mice: H&E stained sections of livers and lipid-droplet count per mm² analyzed by Visiopharm.

Fig rev 2.3 mRNA expression of markers encoding for fibrosis-associated genes or macrophage-markers in livers of HFD-AL and HFD-IF mice.

4. The analysis of scRNAseq data is relatively superficial. Differential gene expression should be performed for each major cell type (such as adipocyte, FAP) and major immune cell types (such as LAM) among different treatment groups.

We agree that differential gene expression in cell types is an important information. As mentioned above the DE analyses comparing HFD-AL with HFD-IF and HFD-IF with HFD-IF-KO groups for all major cell clusters are now provided as **Supplementary Table 1 and 2**. Overrepresentation analysis of de-repressed genes in the adipocyte cluster (**Fig rev 2.1 = new Fig. 6a**) supported our observation of increased catabolic state upon p53 KO.

The expression of genes/protein listed in Figures 1, 2, and 4 should be checked in snRNAseq.

The expression of genes listed in Figures 1, 2, and 4 that were detected in the snRNA-seq dataset are now shown below as stacked violin plots comparing HFD-IF and HFD-IF-KO groups in the adipocyte and LAM cluster (**Fig rev 2.4**). Dominant expression of LAM markers in the HFD-IF group over the other groups is in line with data in Fig 4 and Supplementary Fig 4 of the manuscript.

Fig rev 2.4. Stacked violin plots showing the expression of selected genes in the adipocyte or immune cell cluster in eWAT of HFD-AL, HFD-IF, and HFD-IF-KO mice.

Based on the single nuclei RNA-seq dataset, the expression of p53 and target genes seems to be unchanged among the groups in cell types other than adipocytes. However, due to limitations in the sequencing depth of single nuclei studies, we could detect Trp53 and Cdkn1a only in a low number of cells (see below **Fig rev 2.5**).

Fig rev 2.5. Feature plots showing the expression of Tp53, Mdm2 and Cdkn1a in eWAT of HFD-AL, HFD-IF and HFD-IF-KO mice.

Is the expression of Tp53 and relevant genes altered in other cell types?

To check whether p53 expression is changed in non-adipocytes, we analysed expression levels of p53 and Cdkn1a in a new *ad libitum* fed and intermittent fasted mouse cohort where we also isolated SVF. In contrast to the induction of Trp53 and target genes in intermittent fasted adipocytes (Fig. 2), in the SVF fraction (where we mostly expect immune along with endothelial and progenitor cells) we found no alterations in the expression of *Trp53* and *Cdkn1a* in response to intermittent fasting (**Fig rev 2.6 and new Supplementary Fig. 2b**).

Fig rev 2.6 = new Supplementary Fig. 2b. mRNA expression of *Trp53* and *Cdkn1a* in the SVF isolated from eWAT of HFD-AL and HFD-IF mice.

5. Please double check the label of the color scale in Figure 3F.
The authors thank the reviewer for this catch. We corrected the Figure.

Reviewer #3 (Remarks to the Author):

The current manuscript identifies that P53 in adipocytes plays a key role in induced metabolic remodeling in response to intermittent fasting (IF). The authors observed that IF induces macrophage infiltration in adipose tissues, coinciding with an increase in the expression of apoptosis genes, including P53 in adipocytes. snRNA-seq analysis reveals that IF has a major impact on adipose cell composition. While the number of adipocytes is greatly diminished, the number of immune cells, particularly lipid-associated macrophages (LAMs), is increased. This effect was largely reversed by adipocyte-specific deletion of p53. Adipocyte-specific p53 KO mice showed improved metabolic flexibility in response to IF. The overall studies are well-conducted, and the mouse genetics data are very convincing. The use of snRNA-seq of whole fat is also appropriate. I address the following points to strengthen the manuscript.

1. One of the key observations of this study is that P53 KO adipocytes show increased catabolic responses rather than undergoing apoptosis. It will be important whether P53 has cell-autonomous regulation in lipolysis and β -oxidation. Do the DEGs or KEGG pathway involved in lipid metabolism emerge in p53 KO adipocytes?

We agree that the shift to catabolic metabolism in adipocytes without p53 is an intriguing aspect and thank the reviewer for the suggestion to interrogate DE genes in adipocytes with and without p53 knock-out (now given in **Supplementary Table 1 and 2**). In fact, performing overrepresentation analysis on the genes with increased expression in clusters designated as adipocytes in our snRNA-seq data yielded a clear picture of catabolic metabolic GO biological processes (see **Fig rev 3.1 = new Fig. 6a**).

Fig rev 3.1 = new Fig. 6a. Differentially upregulated genes (IF-KO vs IF) in the adipocyte annotated cell cluster from snRNAseq data was mapped to gene ontology biological process non-redundant (GOBP) using Webgestalt. Enrichment ratio is given on the x-axis and FDR values are given in bars.

This is in line with our qPCR results on increased lipolytic gene expression (original Fig 5G, now Fig 6b) and with new measurements of induced genes involved in fatty acid oxidation (**Fig rev 3.2 and new Fig 6h**) in adipocyte-rich fraction isolated from HDF-IF-KO vs HDF-IF mice.

Fig rev 3.2 = new Fig 6h. mRNA expression of genes involved in fatty acid oxidation in the adipocyte-rich fraction isolated from eWAT of HFD-AL and HFD-IF mice.

This was further supported by new RNAseq results from starved, cultured human adipocytes yielding OXPHOS and fatty acid metabolism pathways as positively enriched after p53 knockdown upon the top 10 GSEA hallmark gene sets (**Fig rev 3.3 and new Supplementary Fig 6a, b**).

Fig rev 3.3 = new Supplementary Fig 6a, b. GSEA hallmark analysis of RNA-seq data from starved, differentiated SGBS adipocytes with siRNA-mediated knock down of p53 (siTP53) compared to scrambled siRNA control (siCtrl). NES, normalized enrichment score; FDR, false positive rate.

Altogether these new data from the snRNA-seq adipocyte cluster, the adipocyte-rich fraction from eWAT, and from cultured adipocytes indicate a p53-dependent, adipocyte-autonomous regulation of fatty acid oxidation, coinciding with the upregulation of lipolytic genes by p53 knock-out as already reported in the initial manuscript.

2. Human data from P53 variants are very intriguing. However, it needs to be addressed whether the two variants have a correlation with P53 expression levels.

The authors thank the reviewer for this comment. To address this question, we analysed TP53 mRNA expression levels in PBMCs (only available biosource of this cohort) from our FMD cohort comparing individuals with the P72 or R72 variant (**Fig rev 3.4 and Supplementary Fig. 7a**). Furthermore, we performed SNP predictions on the RNA sequencing dataset of the adipose tissue collected during bariatric surgery (**Fig rev 3.5**). In both datasets, the p53 variant did not have a significant impact on p53 mRNA expression levels. However, in the bariatric surgery cohort canonical target genes show a trend to increase for the R72 variant, suggesting that p53 downstream signalling might be affected. Notably, due to limitations in the sequencing depth we were not able to predict p53 SNPs from all individuals of the bariatric surgery cohort, which limits the power of this analysis and prevents the stratification of the entire cohort.

These data are in line with previous studies showing that this polymorphism alters p53 downstream signalling without affecting p53 expression levels itself (PMID: 26947067, PMID: 28475405).

Fig rev 3.4 = new Supplementary Fig. 7a. TP53 mRNA was measured with qPCR to compare P72 and R72 variants in PBMCs from the FMD cohort (original Fig 6).

Fig rev 3.5. TPM counts are shown for samples that achieved sufficient sequencing depth to perform SNP calling (subcutaneous WAT of the responder group), with no significant difference in TP53 expression.

- Additional analysis needs to be done to better understand adipose histology.
 - Adipocyte number and size analysis need to be conducted in control, HFD-AL, HFD-IF, and HFD-IF-KO WAT.

Fig 2A and supplementary Fig 5A in the original manuscript, shows that size distribution for adipocytes in eWAT is not affected by IF and p53 knock-out. Tissue weights, reflecting adipocyte numbers, were given in Fig 1C and Fig 5C showing that eWAT weight is significantly affected by diet, but not by adipocyte-specific p53 knock out. Below we provide adipocyte size distribution from a new analysis done in all 4 groups as requested by this reviewer (**Fig rev 3.6 and new Supplementary Fig. S5a, b**) indicating a shift from smaller adipocytes to larger adipocytes in HFD group vs chow group. This trend showed a tendency to be reversed in the fasting groups.

Fig rev 3.6 = new Supplementary Fig. S5a, b **a** Histogram showing adipocyte size distribution as quantified from whole tissue scans ($n=4-6$) using a customized VisioPharm pipeline in chow, HFD-AL, HFD-IF, and HFD-IF-KO groups. **b** Representative images from the 4 groups. Right panels: H&E staining. Left panels: Quantification results showing adipocytes with different sizes in shades of blue according to the categories in a (scale bar = 100 μm).

- Is the expression of P53 in adipocytes a reversible process?

We thank the reviewer for this important question, that we address with additional experiments:

- We performed in vitro refeeding experiments in mouse C3H10T1/2 starved adipocytes, showing that starvation-mediated p53 protein and target gene induction is reversed by increasing time of refeeding (**Fig rev 3.7 = new Supplementary Fig 2g, Fig rev 3.8= new Supplementary Fig 2h**).
- We also measure p53 target gene expression in human differentiated adipocytes (hMADs) after 24h starvation and 4h refeeding (**Figure rev 3.9**).

Importantly, whereas p53 mRNA levels are not changed by starvation or refeeding conditions, p53 protein levels are increased in response to nutrient fluctuations and reduced to basal levels within the first few hours of refeeding. This data is in line with

previous studies showing that the activity of p53 is mainly controlled through post-transcriptional regulations (PMID: 20932800).

Fig rev 3.7 = new Supplementary Fig 2g. Differentiated mature C3H10T1/2 adipocytes which were starved (STV; HBSS/HEPES) for 24 hours, refeed for the indicated times or kept in full medium for the same time (Ctrl). p53 antibody (D2H9O, Cell Signaling) was used to probe blots. GAPDH serves as loading control.

Fig rev 3.8 = new Supplementary Fig 2h. Differentiated mature C3H10T1/2 adipocytes were starved (STV; HBSS/HEPES) for 24 hours and refeed for the indicated times. Canonical p53 target gene expression was measured by qPCR.

Fig rev 3.9 Differentiated mature hMADs were starved (STV; HBSS/HEPES) for 24 hours or refeed for 4 hours. Canonical p53 target gene expression was measured by qPCR.

How is the P53 level affected in the stromal vascular fraction affected by IF?

We measured p53 expression in the SVF fraction isolated from a new HFD cohort, showing no significant change through intermittent fasting (**Fig rev 3.10 = new Supplementary Fig 2b**).

Fig rev 3.10 = new Supplementary Fig 2b. mRNA expression of Trp53 in the SVF isolated from eWAT of HFD-AL, HFD-IF (after the last 24 hours of fasting).

Together, these new data show that starvation-induced activation of adipocyte p53 signalling is reversible by refeeding and that p53 expression is not changed by IF in the SVF.

4. Cell type interaction analysis from snRNA-seq data also showed a profound effect on the interaction between immune cells and endothelial cells. Do they have potential role in P53-dependent remodeling?

Our ligand-receptor analysis (Figure 4a) does not indicate a change in interaction between endothelial cells and adipocytes or endothelial and immune cells when comparing IF and IF-KO groups. In fact, the interaction strength between endothelial and immune cells is rather weak. The analysis indeed showed a strong interaction between adipocytes and endothelial cells, but in contrast to the signalling between adipocytes and immune cells, the interaction strength of adipocytes and endothelial cells was not reduced by adipocyte-specific deletion of p53. This suggests that the immunomodulatory role of endothelial cells in AT (PMID: 37749386), possibly affected by fasting, is unlikely directly mediated by p53 signalling in adipocytes. However, to provide further information on potential secondary effects of endothelial cells in p53-dependent adipose tissue remodelling, we now provide a full list of differentially expressed genes in endothelial cells comparing eWAT of HFD-IF and HFD-IF-KO mice (**Supplemental table 2**).

REVIEWERS' COMMENTS

Reviewer #1 (Remarks to the Author):

The authors have addressed all my comments

Reviewer #2 (Remarks to the Author):

I have no more comments.

Reviewer #3 (Remarks to the Author):

The comments that were addressed well; I have no further remarks.